# HiFC: High-efficiency Flash-based KV Cache Swapping for Scaling LLM Inference

**Inho Jeong**[1,2]∗  **Sunghyeon Woo**[1,3]∗  **Sol Namkung**[1]  **Dongsuk Jeon**[1]†
[1]Seoul National University   [2]SK hynix   [3]NAVER Cloud

## Abstract

Large-language-model inference with long contexts often produces key–value (KV) caches whose footprint exceeds the capacity of high-bandwidth memory on a GPU. Prior LLM inference frameworks such as vLLM mitigate this pressure by swapping KV cache pages to host DRAM. However, the high cost of large DRAM pools makes this solution economically unattractive. Although offloading to SSDs can be a cost-effective way to expand memory capacity relative to DRAM, conventional frameworks such as FlexGen experience a substantial throughput drop since the data path that routes SSD traffic through CPU to GPU is severely bandwidth-constrained. To overcome these limitations, we introduce **HiFC**, a novel DRAM-free swapping scheme that enables direct access to SSD-resident memory with low latency and high effective bandwidth. HiFC stores KV pages in pseudo-SLC (pSLC) regions of commodity NVMe SSDs, sustaining high throughput under sequential I/O and improving write endurance by up to $8\times$. Leveraging GPU Direct Storage, HiFC enables direct transfers between SSD and GPU, bypassing host DRAM and alleviating PCIe bottlenecks. HiFC employs fine-grained block mapping to confine writes to high-performance pSLC zones, stabilizing latency and throughput under load. HiFC achieves inference throughput comparable to DRAM-based swapping under diverse long-context workloads, such as NarrativeQA, while significantly lowering the memory expansion cost of a GPU server system by $4.5\times$ over three years.

## 1   Introduction

Large language models (LLMs) have demonstrated state-of-the-art performance across a wide range of tasks, including machine translation, code generation, and open-domain question answering [1, 2, 3, 4, 5]. However, LLM services are often hindered by significant memory requirements due to the storage of key–value (KV) caches [6, 7, 8, 9]. The size of this KV cache grows proportionally with both the number of layers and the input sequence length, leading to substantial memory usage in state-of-the-art LLMs. This issue becomes particularly pronounced in long-context scenarios, where the cumulative KV cache can easily exceed the capacity of high-bandwidth memory (HBM) on a GPU, resulting in out-of-memory errors or degraded throughput.

To overcome GPU HBM capacity limits, several LLM inference frameworks [7, 10, 11, 12] swap or offload KV cache blocks to host DRAM. However, provisioning and maintaining large DRAM pools requires substantial capital investment, increases power consumption, and demands more intensive cooling, ultimately inflating the cost for long-context inference deployments.

Attempts to leverage lower-cost, higher-capacity NVMe SSDs for KV cache offloading have also been explored recently [7, 10, 11, 13]. However, since GPUs lack a native NVMe interface, every

---

∗Equal contribution.
†Corresponding author.

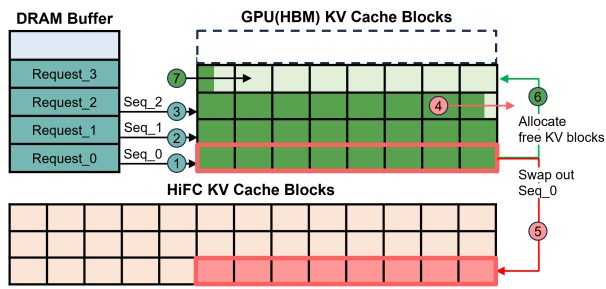
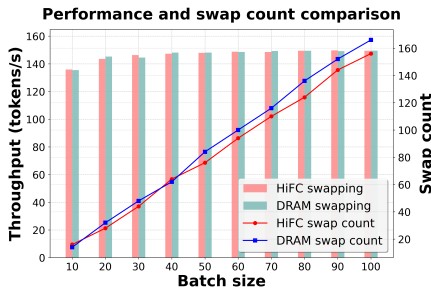

(a) HiFC handles GPU-to-Flash KV cache eviction and prefetch operations without involving DRAM, enabling efficient block-level swap management during decoding.

(b) Comparisons of throughput and total swap counts between HiFC and DRAM swapping systems under increasing batch sizes.

Figure 1: **HiFC workflow and performance. (a) HiFC decoding workflow:** (1)–(3) prompts are tokenized and sequences are allocated KV-cache blocks in HBM; (4) decoding produces additional KV entries; (5) under HBM memory pressure, the scheduler selects a victim sequence and swaps the victim sequence KV cache from HBM to Flash cache (FC); (6) the freed GPU KV blocks are immediately reallocated to the next active sequence; (7) decoding of other sequences proceeds without blocking. **(b) Comparison of throughput and total swap counts between HiFC and DRAM** swapping systems under increasing batch sizes, showing that HiFC achieves comparable performance to DRAM swapping.

Flash I/O transaction must traverse host DRAM and the PCIe fabric, creating a severe bandwidth bottleneck [13]. Moreover, frequent, non-sequential writes increase Write Amplification (WA) [14], which severely accelerates Flash wear and undermines device endurance under sustained operations.

In this paper, we propose **HiFC (High-efficiency Flash Cache)**, a novel DRAM-free paradigm for LLM inference that reduces the memory expansion cost without sacrificing inference throughput. HiFC efficiently offloads the large volume of KV cache generated during LLM inference to Flash-based storage, enabling effective reclamation of GPU HBM and improving sequence-level parallelism. Fig. 1a sequentially illustrates how HiFC is utilized when the HBM capacity is insufficient during the decoding phase. Specifically, HiFC leverages pseudo-SLC (pSLC) regions in Flash to deliver stable high throughput under sequential I/O workloads, while simultaneously improving write endurance, which is measured in total bytes written (TBW), by up to $8\times$. HiFC further employs GPU Direct Storage (GDS) [15] to bypass host memory and eliminate PCIe bottlenecks. Finally, HiFC applies fine-grained block mapping to confine writes to high-performance SLC zones. The proposed DRAM-free architecture achieves comparable throughput even when batch size increases (Fig. 1b) while reducing memory expansion cost by $4.5\times$, providing a scalable solution for demanding long-context inference benchmarks such as LongBench [16].

The main contributions of this work are as follows:

1. **DRAM-Free Design.** We show that large-scale LLM inference can be executed without relying on host DRAM buffers for KV cache swapping, significantly reducing both upfront memory provisioning costs and sustained system-level power consumption.

2. **Optimized GPU-SSD Direct Swapping.** By leveraging GDS and pSLC optimizations, we enable direct and efficient data transfers between a GPU and a Flash-based KV cache, bypassing the bottlenecks of host CPU and DRAM-based buffering.

3. **Seamless Integration with vLLM Framework.** We demonstrate the successful integration of HiFC into the vLLM inference engine. Experimental results confirm that our method can be adopted in real-world frameworks without requiring architectural changes or compromising system stability, achieving a $4.5\times$ lower memory expansion cost and accommodating $4\times$ more requests without performance degradation compared to DRAM-based swapping.

## 2   Background

### 2.1   Importance of KV Cache Management in LLM Service

Inference in decoder-based LLMs consists of two distinct phases: the prefill phase and the decode phase [17]. In the prefill phase, the model processes the entire input prompt in parallel. At each self-attention layer, the key and value vectors for all prompt tokens are stored, denoted as the KV cache. In the decode phase, the model generates one token at a time. For each decoding step, cached key and value vectors from prior tokens are reused to compute self-attention, enabling efficient auto-regressive generation without reprocessing the full sequence at every step.

While this cache-based mechanism significantly improves computational efficiency during decoding, a substantial amount of memory is required to store the KV cache. Specifically, the memory footprint of the KV cache scales proportionally with the batch size, hidden size of the model, and the input sequence length. As a result, scenarios involving large batch sizes or long input contexts can lead to excessive memory usage and potential out-of-memory failures. For instance, serving the DeepSeek-R1 model [5] with merely a few hundred tokens per request can consume hundreds of gigabytes of GPU memory. Therefore, efficient KV cache management is critical for enabling scalable and high-throughput LLM inference, particularly in long-context or multi-request serving environments.

### 2.2   Existing KV Cache Management Techniques

Early LLM serving systems managed the KV cache by pre-allocating a single contiguous GPU buffer sized for maximum context length of each request. Since real sequences are usually shorter, this approach suffers severe internal fragmentation; empirical reports show that only 20–40% of the reserved KV cache space contains useful data, while the rest is wasted [6]. Two complementary techniques have emerged to overcome this limitation:

**Offloading** moves model weights or KV tensors out of GPU HBM to lower-tier devices (e.g., host DRAM or SSD) and keeps them there for many decoding steps [7, 10, 18, 19, 20, 21, 22]. FlexGen [7] exemplifies this strategy: it treats GPU, CPU, and NVMe storage as a unified hierarchy, offloads seldom-used layers or KV blocks, and compresses them to 4 bits to amortize I/O cost. This allowed a single 16 GB GPU to run a 175B-parameter model at reasonable throughput. The trade-off is higher latency whenever disk access is frequent, which can be unacceptable for interactive workloads.

**Swapping**, on the other hand, evicts and restores data on demand in small pages or sequence-level blocks [6, 23, 24]. For instance, PagedAttention in vLLM [6] splits KV cache of each request into uniform blocks of 16–128 tokens. When HBM fills up, selected blocks are temporarily moved to DRAM and fetched back as soon as the request resumes. Uniform page size drives internal and external fragmentation below 5%, and the scheduler caps the swapped volume at the HBM cache size to bound latency. Consequently, vLLM can serve many concurrent requests with long contexts while maintaining low tail latency. In addition, vLLM integrates PagedAttention with block-level KV virtualization and a BlockTable to eliminate fragmentation [10]. During decoding, an LRU-plus priority policy selects victim sequences for asynchronous swap-out to DRAM without pausing GPU execution [25]. A prefetch queue performs swap-in with a single synchronization before kernel launch, enabling a pipelined Running–Swap-Out–Swap-In design that sustains high tensor utilization and throughput [26]. A broader overview of approaches using Computational Storage Devices (CSDs) [13] and techniques targeting larger-scale scenarios in KV cache management [27, 28] is provided in Appendix A.

## 3   HiFC Architecture

### 3.1   Motivation and Design Challenges of DRAM-Free Systems

As discussed in Section 2.2, offloading KV caches to host DRAM or SSD [6, 7] is a common approach to alleviating GPU memory pressure during long-context LLM inference. While offloading KV cache blocks to host DRAM can mitigate the limited capacity of GPU HBM, this approach incurs high operational cost and remains insufficient for memory-intensive workloads such as long-sequence processing or agentic AI, where a large amount of KV cache is required. Alternatively, offloading to commodity NVMe SSDs can reduce infrastructure cost and provide better memory scalability,

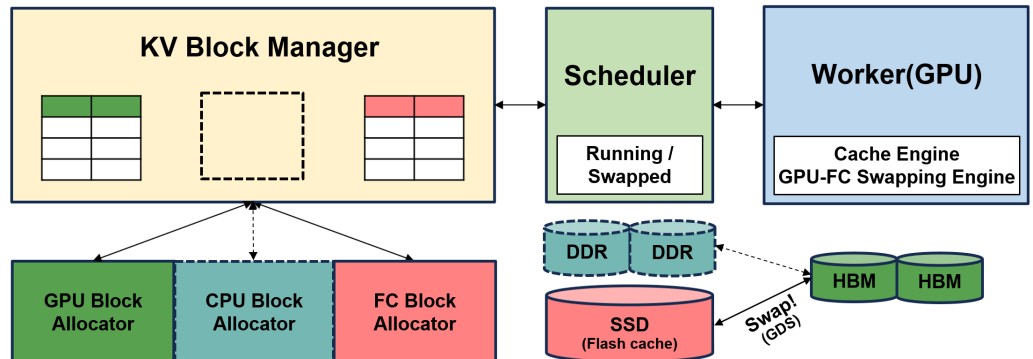

Figure 2: HiFC extends Block Manager of vLLM by integrating a **FC block allocator**, enabling direct GPU-SSD KV cache swapping without relying on host DRAM. The scheduler manages KV block placement based on sequence execution states, and the CUDA-based cache engine leverages GDS to provide an optimized data path between SSD and GPU HBM.

enabling support for larger KV caches and longer contexts. However, this strategy introduces severe throughput degradation, as each I/O must still traverse host DRAM and the PCIe bus. Under bandwidth-limited conditions, this indirect data path significantly impairs decoding performance [13].

These limitations motivate the design of a DRAM-free offloading architecture that bypasses intermediate memory staging and enables direct, high-bandwidth data transfer between GPUs and SSDs. This architecture avoids cumulative latency and bandwidth bottlenecks inherent in existing approaches, while simplifying the overall system design and reducing operational cost.

However, realizing this architecture faces several critical technical challenges. First, SSDs typically offer lower I/O throughput than DRAM, particularly under fragmented or random access patterns, making it difficult to sustain high performance during inference. Second, Flash memory suffers from limited endurance; frequent small writes increase WA and accelerate device wear-out, degrading long-term reliability. Third, traditional offloading paths rely on CPU DRAM as an intermediate buffer, which introduces additional PCIe transfers and redundant memory copies, leading to increased latency and inefficient resource utilization.

### 3.2 HiFC: High-efficiency DRAM-Free Flash Cache Architecture

In Section 3.1, we analyzed the necessity and challenges of building DRAM-free systems. In this section, we propose HiFC, a first DRAM-free design that eliminates the need for host DRAM in KV cache management. HiFC enables a scalable and cost-effective DRAM-free inference stack without compromising throughput or reliability. HiFC achieves these improvements through three tightly integrated components: (1) a Flash cache (FC) block allocator for direct SSD-based KV cache allocation, (2) a Flash-aware block management algorithm to maximize the performance and lifespan of the Flash storage, and (3) a GDS-accelerated cache engine for efficient KV cache swapping between GPU and SSD.

**Flash Cache Block Allocator.** HiFC extends the vLLM memory allocator by introducing FC blocks alongside the existing GPU and CPU blocks. During the decoding phase, when GPU memory is exhausted, existing KV cache blocks are swapped out to FC blocks to free up memory. These FC blocks are managed by the FC block allocator, which applies a block append policy to generate sequential I/O workloads on the SSD. All blocks are of a fixed size (e.g., 32–128 tokens) to maintain compatibility with block-level scheduling of vLLM and minimize fragmentation.

**Flash-Aware Block Management.** Unlike DRAM or GPU memory, SSDs are highly sensitive to random access and block reuse patterns. To address this, the FC block allocator employs a Flash-aware block allocation strategy that ensures the sequential physical placement of KV blocks, thereby reducing internal fragmentation and minimizing WA. Specifically, the module allocates the blocks evicted to Flash in physical order and manages them with a simple append policy that avoids reusing stale logical addresses. This sequential access pattern improves throughput and significantly extends

SSD lifespan [29], achieving a WA factor below 1.02, compared to 1.4 in conventional SSD usage scenarios. These results are validated in Section 5.5.

**GPU–Flash Cache Swapping Engine.**   To maximize throughput and eliminate intermediate buffering, HiFC directly transfers GPU-resident KV tensors to SSD using GDS [15] with byte-level offsets following the scheme in Appendix B for efficient I/O dispatch. To this end, we integrate a Flash cache swapping functionality into the vLLM cache engine. To further enhance I/O efficiency, HiFC leverages multi-threaded I/O scheduling (up to 16 threads) [10] and the direct reuse of tensors as 4KB-aligned GDS buffers [30]. This design sustains a throughput of over 4.7 GiB/s in the pSLC region of SSD, ensuring reliable KV cache swapping operations. As a result, KV swapping with HiFC achieves LLM inference performance comparable to that of DRAM-based approaches, which is validated by our results in Section 5.1.

**HiFC Architecture and Operation.**   This combination of techniques establishes an efficient and scalable DRAM-free memory hierarchy, balancing cost, capacity, and performance for long-context LLM inference. At its core, HiFC replaces the CPU block allocator used by vLLM with a dedicated Flash Cache block allocator, which incorporates Flash-aware block management. This enables a dynamic swapping mechanism: during the decoding stage, when GPU memory becomes scarce, the scheduler identifies a victim sequence group and instructs the worker to initiate a swap-out. The cache engine of worker executes this operation, utilizing a custom CUDA kernel to transfer KV cache data directly to SSD via GDS for maximum throughput. Once sufficient GPU memory is reclaimed, the scheduler directs the worker to swap the swapped sequence group back into GPU memory. As illustrated in Fig. 2, this entire data flow facilitates direct GPU–HiFC KV cache transfers, completely bypassing the host DRAM.

## 4   Memory Expansion Cost Estimation

To evaluate the long-term cost-effectiveness of different KV cache storage mediums, we compute the **memory expansion cost over a 3-year deployment** using Eq. (1), which is based on the Total Cost of Ownership (TCO) model. The cost consists of two components: the capital expenditure (CapEx) required to provision the storage capacity, and the operational expenditure (OpEx) determined by sustained power consumption, datacenter power usage effectiveness (PUE), and regional electricity cost, with all costs denominated in USD.

$$\text{MemExpCost} = \text{CapEx} + \left( \text{Power} \times 24 \times 365 \times \text{Years} \times \text{PUE} \times \frac{\text{EnergyCost}}{1000} \right) \quad (1)$$

Table 1: Three-year memory expansion cost comparison across memory types.

| Metric | DRAM | Enterprise SSD | HiFC (ours) | Saving |
|---|---|---|---|---|
| Capacity | 128 GiB | 1.92 TiB | 1 TiB | – |
| CapEx | $433 | $270 | $118 | 3.6× |
| Power | 64 W | 8.2 W | 5 W | 12.8× |
| 3-year Cost | $614 | $303.6 | $136 | 4.5× |

Table 1 summarizes the 3-year memory expansion cost for three KV cache storage configurations: DRAM (DDR4), enterprise-grade TLC-based SSD, and HiFC (pSLC SSD). The HiFC configuration achieves a 3.6× reduction in capital expenditure and a 12.8× lower power consumption compared to DRAM. This results in a total memory expansion cost of only $136, which is 4.5× lower than the DRAM-based approach. Further details, including projections for future price changes, are provided in Appendix C. Directly utilizing the TLC region of commercial SSDs introduces performance degradation and reduced endurance, leading to cost-inefficiency. HiFC addresses this problem by selecting commercial SSDs that support a pSLC region and operating exclusively within that region, which constitutes 20% of the total capacity in the selected SSDs. This approach extends the drive's lifespan by up to 8× and ensures stable performance. The details of the lifespan are provided in Appendix D.

# 5 Experiments

To ensure consistent and accurate performance benchmarking, we first selected the DeepSeek-R1-Distill-Qwen-32B (DS-Qwen-32B) model [5, 31] and conducted measurements under fixed configurations. To validate the robustness and general applicability of HiFC, we then extended our evaluation to include diverse models and datasets, with these results presented in Section 5.3. In particular, we leveraged the micro-benchmarking utilities provided by the vLLM framework to construct precise execution scenarios and evaluate inference performance quantitatively. The detailed hardware, software, and workload configurations for all experiments are provided in Appendix E.

## 5.1 Performance and KV Cache Swapping Comparison

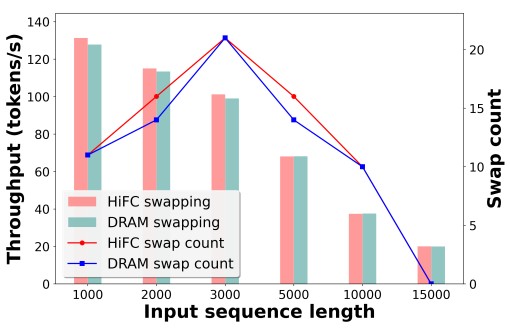 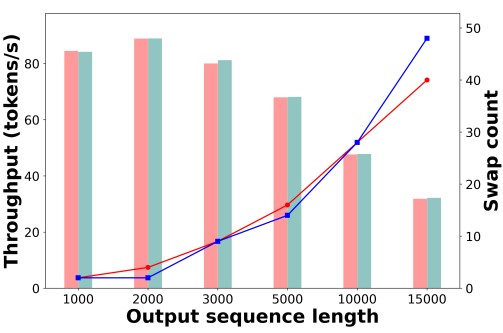

(a) The output sequence length is fixed at 1k.      (b) The input sequence length is fixed at 1k.

Figure 3: **Performance and swap count comparison of HiFC and DRAM swapping** on DS-Qwen-32B with a batch size of 10. (a) As input length grows, throughput for both methods decreases, while swap counts peak around 3k tokens. Both methods show very similar trends. (b) As output length grows, throughput decreases at longer sequence lengths (after 5k), while the swap count steadily increases.

We trigger swap operations by progressively increasing both input and output sequence lengths and compare their impact on system performance. Fig. 3 presents a comparative analysis of KV cache swap performance between HiFC and DRAM under various input and output configurations, where the throughput achieved by both memory types remains comparable. In Fig. 3a, as the input length increases, more KV cache is generated during the prefill phase, but fewer sequences remain active in the decoding stage, reducing effective concurrency and thus throughput. This also leads to fewer swap events.

In contrast, Fig. 3b demonstrates that increasing output length expands KV cache during the decoding phase, allowing for higher concurrency initially. However, as output grows, swap activity surges, while throughput gradually decreases after 5k tokens. Even under this I/O-heavy condition, HiFC achieves near-identical throughput to DRAM, validating its ability to sustain high throughput while maintaining comparable latency, also confirmed by our detailed analysis in Appendix F.

## 5.2 Scalability in Hardware Topologies

To evaluate hardware scalability, we tested HiFC under various GPU-to-SSD ratios. In multi-GPU setups, the pSLC space for the KV cache is partitioned and assigned evenly to each GPU, mirroring the cache management strategy in vLLM. This approach supports flexible deployment topologies, including:

- **1:1 (GPU:SSD)** - A baseline configuration for standard workstations.
- **N:1** - Multiple GPUs sharing a single SSD, optimizing for cost-efficiency.
- **N:N** - Multiple GPUs paired with multiple SSDs (e.g., in a RAID configuration) to maximize both computational performance and I/O bandwidth.

We evaluated the scalability of HiFC across different GPU-SSD topologies using the DeepSeek-R1-Distill-Llama-8B model on two A100 GPUs. The workload consisted of a batch size of 200 from the

GovReport dataset (8k input length on average) with a fixed 1k output length. The request size was calculated based on an average context length of 9k.

Table 2: Performance across different GPU-SSD topologies.

| Setting | DRAM / pSLC Size | Cached requests | Throughput (tokens/s) |
|---|---|---|---|
| GPU+DRAM | 50 GiB | 51 | 172 |
| GPU+SSD (1:1) | 200 GiB | 206 | 182 |
| GPU+SSD (2:1) | 200 GiB | 206 (103 + 103)* | 367 |
| GPU+SSD (2:2) | 400 GiB | 412 (206 + 206)* | 364 |

(*) Using Data Parallelism, each GPU processes an equal number of requests.

The results in Table 2 demonstrate strong scaling capabilities of HiFC. First, it was observed that the HiFC configuration with GPU:SSD = 1:1 achieves comparable performance to the GPU+DRAM baseline. We then scaled the system using Data Parallelism (DP). In 2:1 configuration, the batch was split across both GPUs, achieving a $2\times$ performance increase relative to the baseline. Notably, no I/O bottleneck was observed. Finally, the 2:2 configuration, also operating in the DP mode, maintained this $2\times$ performance gain while simultaneously doubling the total cached request capacity. This N:N setup similarly demonstrated no I/O bottlenecks, confirming HiFC's ability to scale efficiently.

## 5.3 Robustness across Diverse Models and Datasets

Beyond hardware scaling, we validated performance of HiFC in more realistic scenarios. While our primary experiments used a fixed model and context length for fair comparison, real-world deployments must handle diverse workloads. We therefore conducted additional tests using multiple popular open-source models (DeepSeek-Llama-8B [32], DeepSeek-Qwen-14B [33], and Mistral-7B [34]) and datasets with context lengths ranging from a few hundred to over 18k tokens.

As shown in Table 3, HiFC maintains performance similar to DRAM-based swapping across all tested models and datasets. The measured throughput difference remains negligible, with HiFC performing within 1–2% of DRAM, and in some cases even outperforming it. These results underscore the reliability and efficiency of HiFC in practical, diverse inference environments.

Table 3: Measured throughput (tokens/s) on diverse models and datasets.

| Model | KV Swap | Qasper (avg. 3.6k length) | GovReport (avg. 8.7k length) | NarrativeQA (avg. 18.4k length) |
|---|---|---|---|---|
| DS-Llama-8B | DRAM | 302.0 | 172.0 | 95.1 |
| | HiFC | 301.3 | 182.3 | 95.8 |
| DS-Qwen-14B | DRAM | 176.4 | 100.9 | 46.4 |
| | HiFC | 175.3 | 103.5 | 46.6 |
| Mistral-7B | DRAM | 281.0 | 163.0 | 416.1* |
| | HiFC | 275.2 | 162.0 | 406.8* |

(*) Some requests truncated due to Mistral-7B's 32k context-length limit.

## 5.4 Impacts of KV Swapping on Throughput

Fig. 4 illustrates how the vLLM scheduler determines swap events based on the available GPU KV cache budget. We fixed the batch size to 20 and increased the number of concurrently pipelined sequences to examine the emergence and impact of swap behavior. Experiments were conducted on an NVIDIA A100 80 GiB GPU, where 90% of the memory (71 GiB) was allocated: 61 GiB to model weights, 0.97 GiB to activation buffers, and the remaining 9.1 GiB to KV cache.

Based on our theoretical model, when the context length is 5k tokens, the total KV cache footprint exceeds 9.1 GiB beyond 8 concurrent sequences, triggering Flash swap operations. As shown in Fig. 4, swap events for both HiFC and the DRAM baseline begin precisely at this threshold. Although swap counts increase with additional sequences, throughput in both systems continues to increase

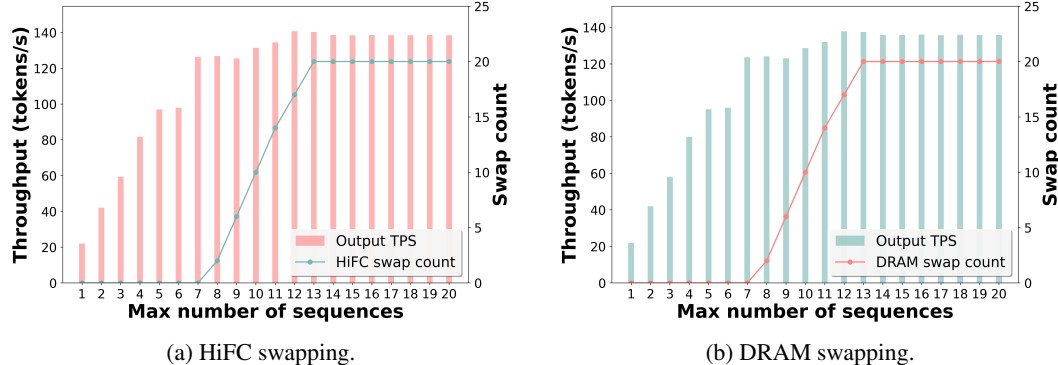

(a) HiFC swapping.

(b) DRAM swapping.

Figure 4: **Performance and swap count trend of HiFC and DRAM swapping.** Each sub-figure shows the average output throughput and the number of swapping event as the max number of sequences increases for the DS-Qwen-32B model, with context length = 5k, batch size = 20, block size = 32, and 100 GiB of dedicated swap space for KV cache for both methods.

before saturating at 12 sequences. This indicates that the swap overhead is effectively absorbed, demonstrating that using HiFC does not negatively impact inference performance compared to DRAM, as both methods become compute-bound, instead of I/O-bound.

This observation is explained by the increasing slack introduced by deeper sequence pipelining. Each additional sequence widens the pipeline's scheduling window, enabling the system to overlap HiFC I/O latency with computation. Appendix G analyzes this effect in detail, showing how pipeline-induced throughput gaps provide sufficient stall budget to hide swap delays, resulting in stable performance even under high swap frequency.

### 5.5 Validating KV Block Management

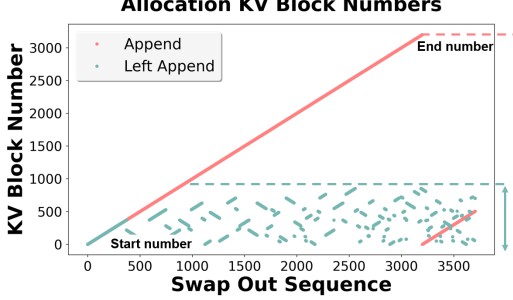

Figure 5: Allocation patterns of KV blocks during swap-out operations under different append strategies.

Table 4: Achieving near-ideal WAF with HiFC swapping.

| Metric | Value | WAF |
|---|---|---|
| Swap out | +232.5 GiB | – |
| Data Units Written | +487,595 | **1.02** |
| SSD Specification | – | 1.40 |
| HiFC vs. SSD Spec. | – | -27% |

Fig. 5 shows the index distribution of FC KV blocks during swap-out operations, illustrating how different free block allocation strategies affect I/O behavior. When using the proposed append strategy, FC KV blocks are allocated in a sequential manner, resembling a typical Least Recently Used (LRU) pattern. In contrast, the left-append strategy yields a markedly different pattern: while the initial 500 swap-out events remain sequential, subsequent allocations become highly non-sequential. This transition reflects a shift from a sequential to a random I/O access pattern.

Furthermore, Table 4 compares the total write volume recorded during swap-out operations with the actual data written to the SSD as reported by the SMART [35] log. In our experiments, 487,595 data units (DUs) were written to the device, equivalent to 232.5 GiB (assuming 512kB per DU). This volume corresponds to a WA factor of 1.02. These findings underscore the importance of sequential block allocation not only for throughput but also for reducing unnecessary Flash wear. The detailed equation is described in Appendix H.

Our evaluation of a commercial SSD quantifies the performance gap between sequential and random I/O workloads. The sequential access patterns achieve significantly higher throughput, confirming that preserving the spatial locality of KV blocks is critical for maximizing HiFC's performance. The performance evaluation results are presented in Section 5.7.

## 5.6 Impacts of Block Size on Swap Latency Hiding

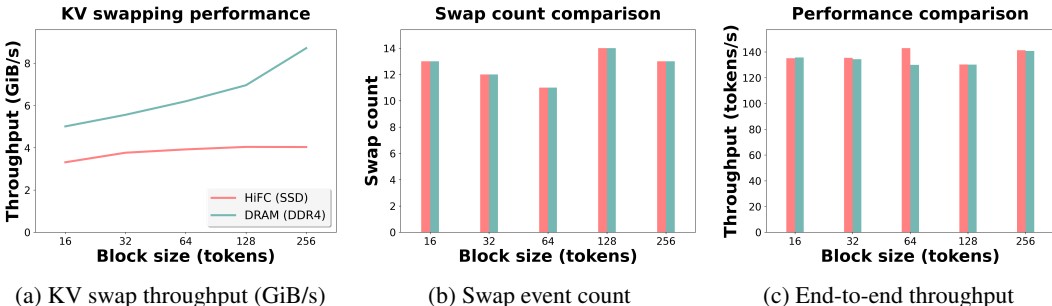

(a) KV swap throughput (GiB/s)  (b) Swap event count  (c) End-to-end throughput

Figure 6: Block size analysis demonstrating that HiFC matches DRAM's end-to-end performance (c), as the raw swap throughput gap (a) is effectively amortized by concurrency.

Fig. 6 compares the impact of block size on KV cache swapping performance for HiFC and DRAM under a long-context workload (context length = 10k, batch size = 10). Fig. 6a shows that both HiFC and DRAM exhibit increasing swap throughput with larger block sizes, where DRAM consistently achieves more than twice the throughput of HiFC. This trend is consistent with prior work such as vLLM [6], which adopts 32-token blocks by default to balance swap cost and fragmentation under paged attention. However, HiFC still achieves nearly identical end-to-end throughput, as shown in Fig. 6c.

As shown in Fig. 6b, the number of swap events is dictated by the workload and block size, resulting in an identical count for both HiFC and DRAM. This finding is crucial as it validates the end-to-end performance comparison in Fig. 6c. We also observe that larger block sizes, such as 128 and 256, can induce more swap events than the 64-token sweet spot, suggesting redundant swapping.

Fig. 6c confirms that end-to-end throughput is improved modestly with larger block sizes, suggesting that coarser-grained swapping has minimal adverse effects on inference performance. This finding aligns with LMCache [25], which also showed that grouping adjacent tokens in physically aligned blocks improves overall throughput without harming latency, especially under decoding workloads.

Overall, these results suggest that the swapping performance gap between HiFC and DRAM does not significantly affect either the swap frequency or end-to-end inference performance, as latency is effectively amortized by concurrent sequence execution. By aligning the block size with the sweet spot observed in Fig. 6b (e.g., 64 tokens), we can maximize swap efficiency without requiring manual tuning for similar workloads.

## 5.7 I/O Throughput Comparisons

Table 5: Sustained GDS I/O throughput across workloads.

| I/O Workload | LBA Range | Min (GiB/s) | Max (GiB/s) | Avg (GiB/s) |
|---|---|---|---|---|
| SEQ WRITE | 100 GiB | 4.341 | 4.724 | 4.715 |
| SEQ READ | 100 GiB | 4.985 | 4.988 | 4.987 |
| RND WRITE | 100 GiB | 1.092 | 2.703 | 1.617 |
| SEQ WRITE | 900 GiB | 1.416 | 1.841 | 1.689 |

Table 5 shows SSD performance benchmarks using the gdsio tool [15] to evaluate the effects of different I/O types and LBA ranges. Sequential reads and writes within the pSLC region delivered consistently high throughput, whereas random writes were significantly slower due to internal

mechanisms such as garbage collection and TLC migration. When the write workload exceeded the pSLC capacity, performance degraded significantly. These findings suggest that structuring KV cache access as sequential I/O is essential for achieving high-performance Flash cache swapping. When the GPU's compute throughput is sufficiently high, the overhead of KV cache swapping can emerge as a bottleneck in the sequence pipelining process.

## 5.8 Scalability of KV Cache Initialization: HiFC vs. DRAM

Table 6: Comparisons of LLM session initialization time.

|  | KV Cache Size (GiB) | | | | | | | | | |
|---|---|---|---|---|---|---|---|---|---|---|
|  | 10 | 20 | 30 | 40 | 50 | 60 | 70 | 80 | 90 | **100** |
| **HiFC (s)** | 33.0 | 34.0 | 33.0 | 33.0 | 33.0 | 33.0 | 33.0 | 33.0 | 32.0 | **32.0** |
| **DRAM (s)** | 39.0 | 46.0 | 46.0 | 59.0 | 59.0 | 60.0 | 89.0 | 89.0 | 90.0 | **90.0** |
| $\Delta$ (= DRAM–HiFC) (s) | 6 | 12 | 13 | 26 | 26 | 27 | 56 | 56 | 58 | **58** |

HiFC significantly reduces the initialization time of LLM inference sessions by eliminating the use of DRAM for KV cache storage. Table 6 reports the LLM session initialization time across different KV cache sizes (ranging from 10 GiB to 100 GiB) for two memory configurations: HiFC and DRAM. HiFC shows a consistently low initialization latency around 32–34 seconds regardless of the KV cache size. In contrast, DRAM-based initialization time increases significantly with cache size, particularly beyond 60 GiB, reaching up to 90 seconds at 100 GiB. Notably, HiFC's performance advantage becomes more pronounced at larger cache sizes: at 100 GiB, the time gap between DRAM and HiFC reaches 58 seconds, resulting in a $2.81\times$ speedup (90.0s vs. 32.0s). This gap arises because HiFC directly utilizes pre-generated Flash cache files, whereas the DRAM baseline must allocate tensors for every KV block upon each initialization. These findings highlight the scalability advantage of Flash-based KV cache initialization, making it a compelling solution for large-batch or long-context LLM inference scenarios.

# 6 Limitation and Future Plan

While HiFC eliminates the need for DRAM and achieves cost-efficient scalability through direct GPU–SSD I/O, several limitations remain. First, in short-context or latency-sensitive workloads, Flash access latency may introduce delays in token processing. Second, in multi-GPU settings, efficient bandwidth scheduling for shared Flash cache becomes increasingly critical to avoid contention. Third, maintaining consistent SSD performance often requires domain-specific expertise in system configuration and filesystem tuning, which may hinder practical deployment. As future work, we plan to explore hybrid Flash–DRAM caching schemes that maintain stable performance across diverse workloads, and to share resources and know-how for SSD optimization.

# 7 Conclusion

In this work, we introduced HiFC, a novel DRAM-free architecture designed to efficiently provide KV cache memory expansion for large-context LLM inference. HiFC replaces conventional DRAM-based swapping with a GDS-enabled SSD utilizing its pseudo-SLC region, thereby creating a direct GPU-to-SSD data transfer pathway that eliminates intermediate host memory bottlenecks. Our experiments demonstrated that HiFC maintains inference throughput comparable to traditional DRAM swapping methods. In addition, our extended validation confirms this advantage holds across diverse models, datasets, and even in multi-GPU scaling scenarios, highlighting HiFC's robustness and general applicability. Furthermore, for long-context and multi-batch scenarios, vLLM's sequence pipelining scheduler and HiFC's optimized Flash cache worked in concert to effectively manage KV cache swaps, thereby minimizing overhead. Simultaneously, substituting 128 GiB of DRAM with a 1 TiB pSLC-configured SSD resulted in an approximately fivefold reduction in the three-year memory expansion cost. Overall, HiFC provides an economically viable and efficient pathway to scalable LLM inference, proving particularly beneficial for large-context and large-batch inference scenarios.

## Acknowledgments and Disclosure of Funding

This work was supported in part by the Institute of Information and Communications Technology Planning and Evaluation under Grant RS-2021-II211343, Grant IITP-2025-RS-2023-00256081, and Grant RS-2024-00347394, and in part by the National Research Foundation of Korea under Grant RS-2024-00408040 and Grant RS-2022-00144419. The authors would like to thank SK hynix for financial support during Inho Jeong's graduate studies and NAVER Cloud for valuable discussions.

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

# A Comparisons with Recent Related Works

## A.1 Comparison to InstInfer

While our main paper discusses related work in the context of general KV cache offloading strategies, this section provides a detailed comparison with InstInfer [13], a concurrent work that also leverages NAND Flash to extend the KV cache. While both HiFC and InstInfer share the goal of moving beyond DRAM, they are built on fundamentally different design principles. InstInfer is a **hardware-specialized solution** that relies on Computational Storage Drive (CSD) to perform in-storage attention processing, thereby alleviating PCIe bottlenecks. In contrast, HiFC is a **general software solution** designed for broad adoption, utilizing commodity NVMe SSDs and GPUs within the popular vLLM framework to maximize cost-efficiency and system compatibility. We summarize the key distinctions across three axes: **cost, performance characteristics, and system compatibility**.

### A.1.1 Cost and Accessibility

The primary difference in cost stems from the required hardware. HiFC is designed to work with ubiquitous, off-the-shelf NVMe SSDs, whereas InstInfer necessitates specialized CSDs. As shown in Table 7, this leads to a significant disparity in initial hardware investment and accessibility, making HiFC a more economically viable solution for immediate, large-scale deployment.

Table 7: Cost and hardware comparison between HiFC and InstInfer.

| Feature | HiFC | InstInfer |
|---|---|---|
| **Required Hardware** | Commodity NVMe SSD | CSD |
| **Example Device** | NVMe Gen4 1TB SSD: ∼$136 | SmartSSD 4TB (used): ∼$2,526 |
| **Implication** | Low cost and easy deployment | Higher initial cost and hardware dependency |

### A.1.2 Latency and Throughput Optimization

HiFC and InstInfer employ different techniques to manage I/O and optimize performance (Table 8). HiFC focuses on hiding I/O latency through vLLM's pipeline-aware scheduling and maximizes bandwidth using GDS, which creates a direct data path between the GPU and SSD. InstInfer, leveraging the computational capabilities of CSDs, processes attention mechanisms directly on the storage device to reduce the volume of data transferred over the PCIe bus. While InstInfer can achieve high throughput by fully exploiting the CSD's internal bandwidth, its performance is tightly coupled to this specialized hardware. HiFC's performance scales with more common components such as SSD bandwidth and GPU DMA engines. A key operational difference is that HiFC swaps KV cache blocks only when HBM capacity is exhausted, whereas in InstInfer's design, the entire KV cache resides in the CSD during inference.

Table 8: Comparison of performance optimization strategies.

| Feature | HiFC | InstInfer |
|---|---|---|
| **I/O Optimization** | GPU Direct Storage (GDS) Pipeline-aware scheduling | In-storage SparF Attention Flash page-level data layout |
| **KV Cache Reduction** | Standard methods (GQA, FP8) | SparF Attention (reduces transfer size) |
| **Performance Gains** | Comparable to DRAM (∼2% diff.) | Up to $11.1\times$ throughput over Flex-Gen (with CSD) |
| **Swapping Behavior** | Swaps only when HBM is full | Entire KV cache resides in CSD |

### A.1.3 System Compatibility and Generality

HiFC is architected as a software-level solution that integrates seamlessly into the widely-used vLLM framework. This approach ensures broad compatibility with existing models, hardware (any GDS-supported NVIDIA GPU and NVMe SSD), and deployment pipelines. InstInfer, being hardware-dependent, requires a more specialized stack, including a modified version of the FlexGen framework and specific drivers for CSDs (Table 9). Consequently, HiFC offers a more general and readily deployable path to scaling inference services, whereas InstInfer provides a powerful but specialized alternative for environments where CSDs are available.

Table 9: Comparison of system compatibility and ease of deployment.

| Feature | HiFC | InstInfer |
|---|---|---|
| **Framework** | Built on vLLM | Modified FlexGen (CSD-specific) |
| **Hardware Req.** | GDS-supported GPU + NVMe SSD | CSD + compatible PCIe infrastructure |
| **Deployment** | Integrates into existing GPU services | Requires specialized hardware setup |
| **Generality** | General-purpose software solution | Specialized hardware-software co-design |

In conclusion, while InstInfer demonstrates excellent performance through hardware acceleration, its adoption is constrained by the cost and availability of specialized CSDs. HiFC, in contrast, delivers a cost-effective, hardware-agnostic, and easily deployable software solution that makes large-context LLM inference practical for a wide range of existing and future systems.

## A.2 Comparison with Other Offloading Systems: AttentionStore and Mooncake

Beyond InstInfer, other recent works such as AttentionStore [36] and Mooncake [27] also address KV cache scaling by offloading to external memory. While sharing the high-level objective of efficient KV cache management, their architectural approaches and target use-cases differ significantly from those of HiFC.

AttentionStore proposes a DRAM/SSD caching hierarchy optimized for multi-turn conversational workloads, where DRAM serves as the primary cache to minimize latency-sensitive SSD access. Mooncake introduces a disaggregated architecture for KV cache sharing across a cluster, primarily targeting GPU efficiency in large-batch, offline inference scenarios.

In contrast, HiFC is designed as a **DRAM-free** online inference solution. It leverages GPU Direct Storage (GDS) to create a direct data path between the GPU and a commodity SSD, a design choice that maximizes I/O bandwidth and fully utilizes the SSD's capacity. By tightly integrating with vLLM's pipeline scheduler, HiFC effectively hides the I/O latency of swapping, achieving DRAM-comparable performance at a fraction of the cost. Table 10 summarizes the key architectural differences.

Table 10: Architectural comparison with AttentionStore and Mooncake.

| Feature | AttentionStore | Mooncake | HiFC (ours) |
|---|---|---|---|
| **Target Workload** | Multi-turn conversation, online service | Long-context, large-batch, offline service | Long-context, large-batch, online service |
| **Memory Hierarchy** | GPUs–[DRAM–SSDs] | Multi-node: N × (GPU–DRAM–SSD) | GPUs–SSDs (DRAM-free) |
| **Latency Handling** | Uses DRAM as a primary cache to minimize I/O overhead from SSD access. | Employs caching and batching to mitigate remote memory access latency. | Uses vLLM's scheduler to hide SSD I/O latency, achieving DRAM-level performance. |

# B  Determining Flash Cache Byte Offsets

The effectiveness of the append FC block allocation strategy is empirically evaluated in Section 5.5, which demonstrates improved sequential I/O behavior under this design. To enable such behavior, the FC byte offset should be carefully determined. While FC blocks are a logical construct for managing the KV cache, actual storage on SSD requires computing the byte-level offset within the Flash cache file. On the GPU, KV blocks are 4KB-aligned tensors. Aligning these logical blocks with the SSD's physical 4KB LBA (Logical Block Address) format is highly beneficial for I/O performance and device lifespan. Therefore, our allocation strategy (HiFC) is designed to enforce this 4KB alignment. To minimize runtime computation overhead, we precompute and store a lookup table of KV-type-specific offsets for each layer. These offsets are then passed directly to the GDS API for I/O dispatch. The equations below represent how those offsets are obtained.

## B.1  Variable Definitions

- $L$: number of attention layers
- $B$: number of FC blocks per KV type per layer
- $H$: number of KV heads
- $S$: block size (tokens)
- $D$: head size
- $T$: data type byte size (e.g., 2 for FP16)

## B.2  Byte Offset Calculation Equations

**Single block size (bytes).**
$$\text{block\_size\_bytes} = H \times S \times D \times T \tag{2}$$

**Layer offset (bytes).**  This is the starting offset for a given layer $l$.
$$\text{layer\_offset} = l \times (2 \times B \times \text{block\_size\_bytes}), \quad \text{where } 0 \leq l < L \tag{3}$$

**Intra-layer KV type offset (bytes).**  This is the relative offset for the Key or Value block region within a layer.
$$\text{key\_type\_offset} = k \times (B \times \text{block\_size\_bytes}), \quad \text{where } k = 0 \tag{4}$$
$$\text{value\_type\_offset} = v \times (B \times \text{block\_size\_bytes}), \quad \text{where } v = 1 \tag{5}$$

**Total byte offset per layer.**  This calculates the final absolute byte offset for the start of the Key/Value cache region for a specific layer.
$$\text{KeyOffset(l)} = \text{layer\_offset} + \text{key\_type\_offset} \tag{6}$$
$$\text{ValueOffset(l)} = \text{layer\_offset} + \text{value\_type\_offset} \tag{7}$$

As a concrete example, we apply the equations above to the DeepSeek-Qwen-32B model with parameters: $L = 64$, $B = 3200$, $H = 8$, $S = 128$, $D = 128$, and $T = 2$. Table 11 shows the computed actual byte offsets (which correspond to KeyOffset(l) and ValueOffset(l)) and their corresponding values in GiB for selected layers.

Table 11: FC byte offsets for selected layers.

| Layer | Key Block Offset (bytes) | Value Block Offset (bytes) | Start Offset (GiB) |
|---|---|---|---|
| 0 | 0 | 838860800 | 0.00 |
| 1 | 1677721600 | 2516582400 | 1.56 |
| 2 | 3355443200 | 4194304000 | 3.12 |
| 62 | 104018739200 | 104857600000 | 96.88 |
| 63 | 105696460800 | 106535321600 | 98.44 |

# C    Memory Expansion Cost Analysis and Future Projections

To substantiate the paper's cost reduction claims, this section provides a detailed memory expansion cost analysis with explicit citations and examines projected market trends to evaluate the long-term economic viability of the HiFC architecture.

## C.1    Updated 3-Year Memory Expansion Cost Analysis (H2 2025)

For enhanced clarity and reproducibility, we have re-evaluated all cost parameters using updated market data from the second half of 2025; the original paper used data from H1 2025. The updated 3-year memory expansion cost comparison is presented in Table 12. All cited prices are derived from publicly available sources to ensure the reproducibility of the analysis (e.g., Memory.NET, Amazon, and the 2024 US Data Center Energy Usage Report).

Table 12: Updated 3-year memory expansion cost comparison based on H2 2025 market data.

| Metric | DRAM (DDR4) | TLC SSD (Gen4) | HiFC (Gen4) |
|---|---|---|---|
| Capacity (GiB) | 128 | 1920 | 1024 |
| Price per GiB ($) | 5.06 | 0.20 | 0.10 |
| CapEx ($) | 647 | 384 | 102 |
| Power Draw (W) | 24 | 8.2 | 5 |
| PUE | 1.3 | 1.3 | 1.3 |
| Energy Cost ($/kWh) | 0.1 | 0.1 | 0.1 |
| **3-year Cost ($)** | **729** | **465** | **120** |
| Previous Comparison | ×4.5 | ×3.9 | 1.0 |
| **Updated Comparison** | **×6.1** | **×3.9** | **1.0** |

The updated analysis reveals that the 3-year memory expansion cost gap has widened, with the DRAM-based solution now being 6.1× more expensive than HiFC (previously 4.5×). This increase is mainly due to recent DRAM price hikes, further strengthening the cost advantage of SSD-based HiFC.

## C.2    Projected Memory Types and Price Trends (2025-2027)

We further analyzed projected price trends for key components to determine if HiFC's cost benefits would shrink, persist, or grow. Table 13 summarizes these projections.

Table 13: Price trends for key hardware components.

| Component | 2025 (Current) | 2027 (Proj.) | Price Trend |
|---|---|---|---|
| GPU (GB200) | $30K–$35K | $20K–$25K | Remains high due to demand |
| DDR5 DRAM (128 GiB) | $1,050–$1,200 | $800–$950 | Decrease as adoption grows |
| Data Center SSD | $350–$420 | $250–$300 | Steady decline with oversupply |
| pSLC SSD (1 TB) | $95–$120 | $75–$100 | Remains cost-effective |
| High-Perf. SSD | $3,000+ | $1,500–$2,000 | Decrease as GDS adoption grows |

The market analysis indicates that HiFC's cost advantage is likely to become even more significant over the coming years. While next-generation GPUs require expensive DDR5 memory, its price is projected to decrease more slowly than that of data center SSDs. HiFC capitalizes on this trend by using cost-effective SSDs for temporary KV cache, avoiding high-cost memory.

The updated market data confirms that HiFC's cost claims remain valid and that its cost benefits are even more pronounced in the current market environment. Future price projections reinforce this conclusion, demonstrating that HiFC provides a practical, economical, and forward-looking solution for large-scale LLM inference deployments.

# D Total Bytes Written (TBW) Enhancement via pSLC SSD

A consumer-grade NVMe Gen4 SSD employs a dynamic pSLC caching mechanism to accelerate write throughput. The total available SLC cache size is approximately 200 GiB. In this experiment, we consider a constrained usage pattern where only a fixed 200 GiB logical block range within the dynamic SLC cache is used exclusively for sequential write and read operations. Since the precise SSD lifetime depends on product-specific warranty policies and variations in internal technology, this analysis uses program/erase (P/E) cycles to provide an approximate lifespan estimation.

**Official Endurance.** According to the datasheet, the SSD-A 1 TB model guarantees 750 TBW (terabytes written) over its lifetime. This figure is based on the assumption of uniform usage across all NAND blocks in TLC mode, with an endurance of approximately 3,000 (P/E) cycles per cell.

**SLC Mode Endurance.** In pseudo-SLC mode, only 1 bit per cell is programmed, which significantly improves endurance to approximately 30,000 P/E cycles per cell:

$$E_{\text{SLC}} = 30,000, \quad E_{\text{TLC}} = 3,000. \tag{8}$$

Let $C_{\text{use}} = 200\,\text{GiB}$ be the capacity of the fixed SLC region used. Then, the theoretical maximum TBW under this scenario is:

$$\text{TBW}_{\text{SLC}} = E_{\text{SLC}} \times C_{\text{use}} = 30,000 \times 200\,\text{GiB} = 6,000\,\text{TiB}. \tag{9}$$

This calculation assumes an ideal WA factor of 1. As demonstrated in Section 5.5, HiFC's sequential append strategy achieves a WA close to 1.0, so it is omitted from this theoretical calculation.

**Comparison to Official TBW.** When compared to the vendor-guaranteed endurance:

$$\frac{\text{TBW}_{\text{SLC}}}{\text{TBW}_{\text{official}}} = \frac{6,000}{750} = 8 \times . \tag{10}$$

This indicates that the SSD can withstand $8\times$ more data writes in an SLC-only usage pattern than under the default TLC-mode assumption, assuming minimal cache folding and ideal sequential I/O behavior.

Table 14: Endurance comparison across different NAND usage modes.

| Usage Mode | P/E Cycles | TBW (TiB) | TBW Multiplier |
|---|---|---|---|
| Official Spec (TLC, full drive) | 3,000 | 750 | $1\times$ |
| Theoretical Max (TLC, full drive) | 3,000 | 3,000 | $4\times$ |
| SLC-only (200 GiB region) | 30,000 | 6,000 | $8\times$ |

These findings confirm that selectively operating within the SLC cache region of the SSD-A significantly improves write endurance. The result is particularly useful for long-lived AI inference caches and KV swap storage systems with highly localized access patterns.

The SLC-Only endurance mode shown in Table 14 is directly incorporated into the HiFC configuration, enabling significantly higher write durability. This configuration is analytically modeled in Eq. (1) of Section 4 and serves as the basis for the memory expansion cost estimation results summarized in Table 1. By leveraging the high-cycle SLC region for KV swap traffic, HiFC achieves sustained endurance gains critical for long-lived inference workloads.

## D.1 Empirical Endurance Analysis and Lifetime Validation

To assess the long-term viability of HiFC, we conducted an endurance test to predict the operational lifetime of the underlying SSD. The primary metric for SSD longevity is Total Bytes Written (TBW), which specifies the total amount of data that can be written to the drive before it is likely to fail.

### D.1.1 Experimental Setup and Lifetime Formula

The endurance test was conducted using the following configuration:

- **GPU:** NVIDIA A100 (80 GiB)
- **SSD for HiFC:** 1TB (pSLC mode, 200 GiB capacity, 6,000 TBW rating)
- **Dataset:** GovReport (avg. 8.7k sequence length)
- **Output length:** 1k tokens

The predicted lifetime in years is calculated based on the drive's rated pSLC TBW and the empirically measured daily write volume, as shown in Equation 11.

$$\text{Lifetime (years)} = \frac{\text{pSLC TBW (TiB)}}{\text{Total KV Written per Day (TiB)} \times 365} \tag{11}$$

### D.1.2 Results and Key Findings

The key metrics derived from the endurance test are presented in Table 15.

Table 15: SSD endurance test results and lifetime prediction under a sustained workload.

| Metric | Value |
|---|---:|
| Swap-out count (writes) | 28 |
| Avg. KV size (MiB) | 984 |
| KV size per test (GiB) | 26 |
| Total test time (s) | 1,097 |
| Tests per day (extrapolated) | 78 |
| Total KV written per day (TiB) | 1.98 |
| pSLC TBW rating (TiB) | 6,000 |
| **Predicted Lifetime (years)** | **8.3** |

The analysis predicts that under our sustained test conditions, the SSD utilized by HiFC would last for approximately 8.3 years. This operational lifespan significantly exceeds the typical expected service life of GPUs used for LLM inference, which is often estimated to be around 3 years. This result implies that for every three GPU replacement cycles, the SSD would only need to be replaced once. Therefore, HiFC not only provides a cost-effective solution for memory expansion but also demonstrates exceptional long-term endurance, ensuring that the storage component does not become a frequent point of failure or replacement in a production environment.

# E Experimental Environments

## E.1 Performance and KV Cache Swapping Test Environment

Table 16: Hardware and software configuration used in our experiments.

| System & Hardware | |
|---|---|
| Server / CPU | Dell PowerEdge R750xa, 2 × Intel Xeon Silver 4310 |
| GPU | 2 × A100 (80 GiB) |
| Memory | 256 GiB DDR4 |
| OS | Ubuntu 22.04.5 LTS |
| System Storage | Samsung PM893, 7.68 TB SATA eSSD |
| Flash cache | SSD, 1 TB NVMe Gen4 (pSLC cache: 200 GiB) |
| **Software & LLM Model** | |
| Baseline | vLLM v0.6.6 |
| CUDA Toolkit | 12.3 |
| PyTorch | 2.5.1 |
| GDS Release | 1.8.1.2 |
| LLM Model | DeepSeek-R1-Distill-Qwen-32B |

## E.2 Diverse Models and Datasets Test Environment

Table 17: Experimental setup for mixed workload validation.

| Item | Specification |
|---|---|
| **Models** | DeepSeek-Llama-8B, DeepSeek-Qwen-14B, Mistral-7B |
| **Datasets** | Qasper (avg. 3.6k), GovReport (avg. 8.7k), NarrativeQA (avg. 18.4k) |
| Block size | 64 tokens |
| Output length | 1k |

## E.3 Validation Environment for WAF

Table 18: Validation environment and test workloads.

| Model & HiFC setting | |
|---|---|
| Model | DeepSeek-R1-Distill-Qwen-32B |
| Max Model Length | 10,000 tokens |
| HiFC Space | 100.0 GiB |
| Number of SSDs | 1 |
| **Test workload** | |
| Input sequence length | 5,000 tokens |
| Ouput sequence length | 5,000 tokens |
| Request | 50 |

# F  Latency Impact Analysis of HiFC Swapping

A critical consideration for swapping-based approaches is the potential introduction of unacceptable I/O latency, which could be problematic in latency-sensitive serving systems. This section evaluates HiFC's swapping mechanism under a heavy request load to quantify its impact on end-to-end throughput and request latency.

As with any swapping mechanism, moving KV cache from GPU HBM to another memory tier introduces some overhead. In vLLM, this is handled through two primary strategies: **Recompute**, which avoids I/O but incurs additional computation, and **Swapping**. To maintain low latency, the vLLM scheduler preemptively sacrifices a few requests to free up HBM, thereby preventing overall service delays. HiFC adopts this same scheduling logic, allowing it to hide most of the swap latency through pipeline overlapping.

## F.1  Experimental Setup

The evaluation was conducted using the following configuration:

- **GPU:** NVIDIA A100 (80 GiB)
- **Model:** DeepSeek-R1-Distill-Llama-8B
- **HBM KV size:** 4 GiB (starvation condition for recompute and swapping)
- **Dataset:** THUDM/LongBench/gov_report_e (avg. 8K sequence length)
- **SSD for HiFC:** 1TB (pSLC 200 GiB)

## F.2  Evaluation Results

We compared HiFC against a GPU-only baseline and a DRAM-based swapping system. The GPU-only configuration quickly ran into Out-of-Memory (OOM) errors under the immense request load, highlighting the necessity of a memory expansion strategy. The performance comparison is detailed in Table 19.

Table 19: Performance comparison of memory management strategies under a heavy, latency-sensitive request load.

| Metric | GPU only Recompute | DRAM-based Swapping | HiFC Swapping |
|---|---|---|---|
| HBM for KV cache (GiB) | 4 | 4 | 4 |
| KV cache size (GiB) | 4 (OOM) | 50 | 200 |
| Output tokens/s | 179 | 172 | 182 |
| Avg. latency (s/request) | 5.5 | 5.8 | 5.4 |

## F.3  Observations and Conclusion

The results in Table 19 shows that HiFC provides superior performance, achieving 5% higher throughput (182 vs. 172 tokens/s) and 7% lower average latency (5.4 s vs. 5.8 s) compared to the DRAM baseline. This is possible because HiFC successfully hides swap latency through vLLM's I/O-computation overlapping and avoids the DRAM contention bottleneck (a source of tail latency) by using a dedicated GDS I/O path. These findings provide strong evidence that HiFC's design is highly effective for latency-sensitive inference, preventing SSDs from becoming a performance bottleneck.

# G   Pipeline-Aware Model of Swap Overhead

**Pipeline-induced throughput gap.**   Under deep pipelining, the realised throughput $\text{TPS}_{\text{test}}$ often falls short of the ideal value $\text{TPS}_{\text{ideal}} = \text{RUN}_{\text{seq}} \cdot \text{TPS}_{\text{single}}$. We define the per-second throughput deficit

$$\Delta_{\text{TPS}} = \text{TPS}_{\text{ideal}} - \text{TPS}_{\text{test}} \quad (\geq 0). \tag{12}$$

Let $T_{\text{lat}}$ be the mean per-token pipeline latency (e.g., 45 ms in our measurements). The deficit translates into a *stall budget* that the pipeline can tolerate without additional loss of throughput:

$$T_{\text{gap}} = \Delta_{\text{TPS}} \, T_{\text{lat}}. \tag{13}$$

**Flash-swap latency.**   Swapping $N_{\text{swap}}$ victim blocks of $B_{\text{size}}$ tokens each incurs

$$T_{\text{swap}} = T_{\text{fix}} + \frac{N_{\text{swap}} \, B_{\text{size}} \, s_{\text{tok}}}{\text{BW}_{\text{Flash}}}, \tag{14}$$

where $T_{\text{fix}}$ is a constant software overhead, $s_{\text{tok}}$ the memory footprint per token, and $\text{BW}_{\text{Flash}}$ the sequential bandwidth of the SSD.

**Visibility criterion.**   Swap overhead is *visible* only if it exhausts the stall budget:

$$\text{visible} \quad \Longleftrightarrow \quad T_{\text{swap}} > T_{\text{gap}} \quad \big(\text{otherwise hidden}\big). \tag{15}$$

**Empirical verification.**   Table 20 summarises a representative run with a 32-MiB block size (128 tokens). Although a 29-block swap adds 266 ms of Flash latency, the pipeline already exhibits a 46 tokens s$^{-1}$ deficit, yielding a $T_{\text{gap}} = 2.07$s (Eq. (13)); inevitably, the swap latency is absorbed and no additional performance drop is observed.

Table 20: Pipeline gap–based swap-visibility analysis.

| Timestamp | $\text{RUN}_{\text{seq}}$ | $\Delta_{\text{TPS}}$ | $T_{\text{gap}}$ (ms) | $T_{\text{swap}}$ (ms) | Eq. (15) | Visible |
|---|---|---|---|---|---|---|
| 22:26:45 | 1 | 0 | 0 | – | false | No |
| 22:27:21 | 7 | 12 | 540 | – | false | No |
| 22:27:45 | 11 | 44 | 1 980 | – | false | No |
| **22:27:51** | **13** | **46** | **2 070** | **266** | false | **Hidden** |

This pipeline-aware model generalises the single-sequence criterion and explains why Flash swapping remains performance-neutral under deep batching and long-context workloads.

# H  Analysis of Write Amplification Factor (WAF)

We analyze the Write Amplification Factor (WAF) of HiFC to evaluate its impact on SSD endurance. WAF is a critical metric for Flash-based storage, defined as the ratio of data physically written to the NAND Flash memory to the data written by the host system. A lower WAF signifies greater I/O efficiency and contributes to a longer device lifespan.

We calculate the empirical WAF using metrics from the device's SMART (Self-Monitoring, Analysis, and Reporting Technology) logs, as summarized in Table 21. The host writes correspond to the "Swap out" volume, while the NAND writes are derived from "Data Units Written," which the device reports in 512 kB units.

Table 21: Host write volume vs. actual NAND write volume during the vLLM swapping test.

| Source | Metric | Value |
|---|---|---|
| *vLLM (Host Writes)* | | |
| | Block Size | 32 MB |
| | Swap Out Count | 7,440 |
| | **Swap Out Size** | **232.5 GiB** |
| *SSD SMART (NAND Writes)* | | |
| | Data Units Written (End)* | 238,896,067 |
| | Data Units Written (Start)* | 238,408,472 |
| | Data Units Written (Diff)* | 487,595 |
| | **Total NAND Write Size** | **232.5 GiB** |

The WAF is computed by dividing the total bytes written to NAND by the total bytes written from the host, as shown in Eq. (16).

$$\text{WAF} = \frac{\text{NAND Writes (B)}}{\text{Host Writes (B)}} = \frac{487,595 \times 512 \times 1,000}{7,440 \times 32 \times 1024^2} \approx 1.02 \tag{16}$$

The resulting WAF of 1.02 is exceptionally low, indicating that for every 1.02 bytes written to the physical NAND Flash, 1.00 byte was requested by the host. This value is 27% lower than the SSD's own specification of 1.4, which represents a typical WAF for random read/write workloads. This demonstrates that the large, sequential write patterns generated by HiFC are highly beneficial for the endurance and longevity of the underlying Flash storage.

