# OpenReview forum: "HiFC: High-efficiency Flash-based KV Cache Swapping for Scaling LLM Inference"
_NeurIPS.cc/2025/Conference — NeurIPS 2025 poster_

### Official Review · Reviewer_Gyax · 2025-07-01

**Clarity:** 2
**Significance:** 2
**Originality:** 3
**Rating:** 4
**Confidence:** 3

**Summary:**

This paper presents HiFC (High-efficiency Flash Cache), an efficient KV cache management solution for large language model inference. Addressing the high cost of DRAM-based swapping and poor performance of conventional SSD offloading, HiFC enables direct GPU-to-SSD data transfer bypassing host DRAM while leveraging SSD's pseudo-SLC region for optimized I/O performance. Experimental results demonstrate comparable inference throughput (TPS) to DRAM solutions with significantly reduced costs. The method seamlessly integrates with existing vLLM frameworks and proves effective for long-context and high-batch inference scenarios.

**Questions:**

Please refer to Weaknesses for details. I would increase the score if these points are adequately addressed.

**Ethical Concerns:**

["NO or VERY MINOR ethics concerns only"]

**Final Justification:**

The paper propose an efficient KV cache management solution for large language model inference. Addressing the high cost of DRAM-based swapping and poor performance of conventional SSD offloading. The rebuttal addressed most of my concerns.

**Limitations:**

yes

**Quality:**

2

**Strengths And Weaknesses:**

Strengths
S1)HiFC provides a practical way to reduce the expensive memory costs of running large language models by effectively using SSDs as extended GPU memory.
S2)The system works smoothly with existing popular frameworks like vLLM without requiring changes to model architectures.

Weaknesses
W1)The paper focuses more on system building than new AI algorithms, so I believe its practical performance should be the most important criterion for evaluating its contributions.  It only tests with one GPU and one SSD, while real applications often use multiple GPUs and SSDs. We don't know how well HiFC would work in these more common setups.
W2)The experiments only use one model (DS-Qwen-32B) and fixed-length inputs (like 5K/10K tokens). In real-world use, LLMs handle mixed workloads with varying context lengths. We don't know if HiFC works as well with different model sizes or dynamic input lengths that change during operation.
W3)The paper could be more convincing if it includes a comparison with InstInfer, especially in terms of cost, latency, and compatibility. Such a comparison would provide a more comprehensive evaluation of the strengths and weaknesses of the proposed method relative to existing solutions.

---

> ### Author Rebuttal · Authors · 2025-07-31
>
> We appreciate the reviewer’s thorough assessment of our work and the constructive suggestions provided. Our detailed responses to the raised points are given below.
>
> **Q4.1.** How does HiFC perform and scale in realistic multi-GPU and multi-SSD deployments?
>
> **A4.1.** We appreciate the reviewer’s insightful comment regarding the scalability of HiFC. As stated in our paper’s limitations, we have extended HiFC to support **multi-GPU and multi-SSD** configurations and confirmed its scalability with new experiments.
>
> HiFC supports the following topologies:
>
> - **1:1 (GPU:SSD)** – baseline configuration for workstation-level applications
> - **N:1** – multiple GPUs sharing one SSD, minimizing SSD cost
> - **N:N** – multiple GPUs with multiple SSDs (RAID-enabled), maximizing both GPU performance and SSD bandwidth
>
> **Experimental Setup**
>
> | Item | Specification |
> | --- | --- |
> | GPU | A100 (80GB) × 2 |
> | HBM KV cache size | 4 GiB (for HBM starvation condition) |
> | DRAM KV cache size | 50 GiB |
> | SSD for HiFC | pSLC 200 GiB × 2 |
> | Model | DeepSeek-R1-Distill-Llama-8B |
> | Dataset | THUDM/LongBench/gov_report_e (avg. 8K length) |
>
> **Extended Experimental Validation**
>
> | Setting | DRAM / pSLC Size | Requests (avg. 8K) | TPS (tokens/s) |
> | --- | --- | --- | --- |
> | GPU:DRAM | 50 GiB | 51 | 172 |
> | GPU:SSD = 1:1 | 200 GiB | 206 | 182 |
> | GPU:SSD = 2:1 | 200 GiB | 206 (103 + 103) | **367** |
> | GPU:SSD = 2:2 | 400 GiB | 412 (206 + 206) | **364** |
>
> **Key Findings:**
>
> - HiFC with a **1:1** configuration already outperforms DRAM-based swapping by **5% TPS** while providing much larger KV cache capacity.
> - **2:1 scaling** achieves **~2× throughput**, as swap overlaps rarely collide on a single SSD.
> - **2:2 scaling** maintains the **2× throughput increase** while expanding request capacity.
> - For multi-GPU setups, the pSLC space for KV cache is evenly assigned to each GPU, consistent with the cache management strategy used in vLLM.
> - Adding more SSDs increases request capacity. Select SSDs according to LLM size.
>
> **Conclusion:**
>
> These results confirm that HiFC scales effectively across multi-GPU and multi-SSD setups, preserving both throughput and memory capacity advantages.
>
> ---
>
> **Q4.2.** Does HiFC maintain its performance across different model sizes and dynamic, mixed-length workloads?
>
> **A4.2.** We appreciate the reviewer’s question regarding HiFC’s performance under mixed workloads and varying model sizes. HiFC was designed with real-world deployment in mind and has been evaluated on diverse datasets during development. The paper used a fixed model and input/output length to ensure a fair comparison with DRAM-based swapping. For additional verification, we conducted new tests using multiple popular open-source models (DeepSeek-Llama-8B, DeepSeek-Qwen-14B, and Mistral-7B) and datasets with **context lengths ranging from a few hundred to over 18K tokens**. These tests were run with the same GPU/SSD configuration as in the paper.
>
> **Experimental Setup**
>
> | Item | Specification |
> | --- | --- |
> | **GPU** | A100 (80GB) |
> | **HBM KV cache size** | 4 GiB (for HBM starvation condition) |
> | **KV swap memory** | DRAM (DDR4) / SSD (GEN4) |
> | **Datasets** | Qasper (avg. 3.6K), GovReport (avg. 8.7K), NarrativeQA (avg. 18.4K) |
> | **Block size** | 64 tokens |
> | **Output length** | 1K |
>
> **Extended Experimental Validation**
>
> | Model | KV Swap | Qasper | GovReport | NarrativeQA | Observation |
> | --- | --- | --- | --- | --- | --- |
> | **DS-Llama-8B** | DRAM | 302.0 | 172.0 | 95.1 |  |
> |  | HiFC | 301.3 | 182.3 | 95.8 | Avg. diff. ~2% |
> | **DS-Qwen-14B** | DRAM | 176.4 | 100.9 | 46.4 |  |
> |  | HiFC | 175.3 | 103.5 | 46.6 | Avg. diff. ~1% |
> | **Mistral-7B** | DRAM | 281.0 | 163.0 | 416.1* | NarrativeQA only <32K |
> |  | HiFC | 275.2 | 162.0 | 406.8* | Avg. diff. ~2% |
>
> **Key Findings**
>
> - Across all tested models and datasets, the **TPS difference between DRAM and HiFC remained within ~2%**.
> - Results are consistent with those reported for DS-Qwen-32B in the paper.
> - Mistral’s 32K limit affected some NarrativeQA requests, but performance trends remained similar.
>
> **Conclusion:**
>
> These additional experiments confirm that **HiFC maintains DRAM-comparable throughput** across diverse models and dynamic context lengths. Therefore, HiFC is robust enough to replace DRAM in long-context LLM inference services under realistic mixed workloads.
>
> ---
>
> **Q4.3.** How does HiFC compare to InstInfer in terms of cost, latency, and compatibility?
>
> **A4.3.** We thank the reviewer for suggesting a comparison with **InstInfer**, an excellent work that leverages Computational Storage Devices (CSDs) to optimize KV cache management and reduce I/O bottlenecks. Both InstInfer and HiFC aim to extend KV cache to NAND Flash, but they adopt fundamentally different approaches:
>
> - **InstInfer** is a **hardware-specialized solution** using CSDs to process KV cache with in-storage attention, eliminating PCIe bottlenecks.
> - **HiFC** is a **software-general solution** built on the widely used **vLLM** framework, utilizing commodity GPUs and SSDs to achieve cost efficiency and compatibility.
>
> The differences in **cost**, **latency/throughput**, and **compatibility** are summarized below.
>
> **Cost Comparison**
>
> | Feature | HiFC | InstInfer |
> | --- | --- | --- |
> | **Required hardware** | Commodity NVMe SSD (pSLC) | Computational Storage Device (CSD) |
> | **Example device** | NVMe GEN4 1TB pSLC SSD: **$136** | SmartSSD(CSD) 4TB (reseller): **$2,526** |
> | **Cost ratio** | **≈18× cheaper**  | Higher initial cost due to specialized hardware |
>
> InstInfer requires CSD hardware (e.g., Daisyplus OpenSSD or SmartSSD), which significantly increases the initial cost. In contrast, HiFC uses readily available pSLC SSDs, resulting in a much lower TCO and easier deployment.
>
> **Latency / Throughput Comparison**
>
> | Feature | HiFC | InstInfer |
> | --- | --- | --- |
> | **I/O Optimization** | GPU Direct Storage (GDS), pSLC SSD focus, pipeline-aware scheduling | In-storage SparF Attention, flash page-level data layout |
> | **KV Cache Compression** | GQA and FP8 quantization **reduce KV cache size by up to 4×** | SparF Attention **reduces KV cache transfer size by up to 8×** |
> | **Performance Dependency** | Scales with **SSD bandwidth** and **GPU DMA performance (GDS)** | Depends on **CSD internal flash bandwidth** and **PCIe P2P DMA efficiency** |
> | **Measured Gains** | Comparable to DRAM with <2% latency difference; TPS scales with SSD/GDS | Up to **11.1×** throughput over FlexGen (with CSD) |
> | **Swapping Behavior** | KV cache is swapped only when HBM is insufficient | All KV cache resides in CSD during inference |
>
> HiFC hides swap latency via pipeline scheduling and uses GDS to minimize CPU/DRAM involvement. InstInfer achieves higher gains when fully utilizing CSD’s internal bandwidth but is more hardware-dependent.
>
> **Compatibility**
>
> | Feature | HiFC | InstInfer |
> | --- | --- | --- |
> | **Framework** | Built on **vLLM** (fully compatible with major models and options) | Modified **FlexGen** (CSD-specific) |
> | **Hardware Requirements** | Works with any **NVMe SSD** on NVIDIA GPUs supporting **GDS** | Requires **CSD** and compatible PCIe switches/drivers |
> | **Deployment** | **Easily integrates** into existing GPU inference services without requiring specialized hardware | **Specialized setup** required |
> | **Generality** | **General solution** for LLM inference service scaling | **Specialized solution** limited to CSD-enabled environments |
>
> InstInfer demonstrates strong scalability but relies on specialized hardware and custom software. HiFC, as a software-level solution, is broadly compatible with existing LLM frameworks and hardware, making it easier to deploy at scale.
>
> **Conclusion**
>
> The additional experiments show that **HiFC achieves throughput on par with DRAM across** diverse LLM architectures and varying context lengths, while reducing the **total cost of ownership by 4.5×** through the use of an inexpensive pSLC SSD instead of costly DRAM. These findings highlight HiFC’s practicality and cost-efficiency, demonstrating its suitability for long-context LLM inference in real-world deployments.
>
> ---
>
> We have implemented these features and will release the full code and results on GitHub before the camera-ready version.
>
> We appreciate the reviewers’ time and consideration in reviewing our rebuttal.

---

> > ### Comment · Reviewer_Gyax · 2025-08-05
> >
> > Thank you for your reply, which addressed most of my concerns. I will increase the score.

---

> > > ### Author Response · Authors · 2025-08-05
> > >
> > > Dear Reviewer Gyax,
> > >
> > > We sincerely thank the reviewer for taking the time to provide such a careful and thoughtful evaluation of our work. The reviewer’s questions offered us an invaluable opportunity to further substantiate that HiFC consistently maintains stable performance across diverse models and long-context datasets, a finding we are pleased to present. We greatly appreciate the insightful comments and constructive suggestions, and we will carefully refine and organize the measured results to incorporate them into the final version of the paper.
> > >
> > > Sincerely,
> > >
> > > Authors of Paper #12452

---

### Official Review · Reviewer_bZj1 · 2025-07-03

**Clarity:** 3
**Significance:** 3
**Originality:** 3
**Rating:** 5
**Confidence:** 3

**Summary:**

This paper presents a flash memory swapping scheme which swaps KV cache pages into an SSD drive. It is designed for LLM inference in memory-constrained environments, where the memory required for KV cache activations is too large to fit in GPU and thus the KV cache needs to be swapped / offloaded to a device with higher memory capacity. They present SSD drive swapping as an alternative to standard methods (offloading to CPU DRAM) which can achieve improved TCO. Their approach uses GPU-Direct-Storage (GDS) to transfer data directly from the GPU to the SSD. It attains similar throughput with DRAM swapping strategies with substantial reduction of TCO relative to providing access to large CPU DRAM pools. They also leverage custom allocation strategies to optimize for reuse in order to enhance device durability by promoting sequential memory accesses.

**Questions:**

- Are the price trends for these technologies (DRAM versus SSD) such that the TCO benefits from this method will shrink, persist, or grow over the coming years?
- Are there any sustainability concerns from repeatedly burning through flash drives (even if this has lower TCO)?

**Ethical Concerns:**

["NO or VERY MINOR ethics concerns only"]

**Final Justification:**

The author rebuttal has addressed each of the concerns I raised as well as the questions I had. As such, I will stick with my original rating (Accept).

**Limitations:**

yes

**Quality:**

4

**Strengths And Weaknesses:**

Strengths:
- They present a DRAM-free strategy with substantial TCO reduction by exploiting flash memory for offloading.
- GDS eliminates the bottleneck of PCIe transfers when offloading to SSD (and allows high I/O transfers)
- Their utilization of single-level cell modes is clever for improving write endurance as well as speeding up I/O transfers (as well as their other optimizations in terms of coalescing memory operations)
- They design a flash-aware allocation strategy to optimize for reuse patterns in order to minimize wear on the SSD devices
- They provide an end-to-end vLLM integration to demonstrate the real-world feasibility of their method
- They present a thorough evaluation of their method, including stress testing their system throughput with different input/output configurations and demonstrating the importance of their sequential access patterns.

Weaknesses:
- This work mainly focuses on throughput-oriented LLM inference. For latency-oriented serving systems, this type of swapping-based method may create unsuitable latency overheads.
- The TCO analysis is not thoroughly justified - there should be citations provided for the price ranges / other components in Table 1. This is particularly critical since claims around reduced power consumption / TCO are central to the aims of the paper (which aims to provide similar performance for reduced cost).
- As mentioned in the limitations section, there are potential systems engineering challenges with maintaining SSD-based systems due to increased failure rate.

---

> ### Author Rebuttal · Authors · 2025-07-31
>
> We appreciate the reviewer’s thorough assessment of our work and the constructive suggestions provided. Our detailed responses to the raised points are given below.
>
> **Q3.1.** Will the swapping-based approach used in HiFC introduce unacceptable latency overheads in latency-oriented serving systems?
>
> **A3.1.** We truly appreciate the reviewer’s insightful question regarding whether HiFC’s swapping approach could introduce problematic latency in latency-sensitive serving systems. This is an important aspect, and we carefully examined it during our evaluation.
>
> We assumed GPU VRAM alone couldn’t handle the immense request load.
>
> As with any swapping mechanism, moving KV cache from GPU HBM to another memory tier introduces some overhead. In **vLLM**, this is handled through two strategies:
>
> - **Swapping**, which adds I/O overhead during KV cache transfer.
> - **Recompute**, which avoids I/O but incurs additional computation.
>
> To keep latency under control, the **vLLM scheduler** selectively sacrifices a few requests to free up HBM, preventing overall service delays. HiFC adopts the same scheduling logic, and our results show that it maintains **end-to-end throughput** and **request latency** comparable to DRAM-based swapping.
>
> **Experimental Setup**
>
> - GPU: A100 (80GiB)
> - HBM KV cache size: 4 GiB (starvation condition)
> - DRAM KV cache size: 50 GiB
> - SSD for HiFC: 1TB (pSLC 200 GiB)
> - Model: DeepSeek-R1-Distill-Llama-8B
> - Dataset: THUDM/LongBench/gov_report_e (avg. 8K length)
>
> **Evaluation Results**
>
> |  | **GPU only** | **DRAM-based** | **HiFC** |
> | --- | --- | --- | --- |
> | **OOM Prevention** | Recompute | Swapping | Swapping |
> | **HBM for KV cache (GiB)** | 4 | 4 | 4 |
> | **KV cache size (GiB)** | 4 | 50 | 200 |
> | **Memory bandwidth (GB/s)** | 1,935 (spec) | 28 (estimated) | 7 (spec) |
> | **Output tokens/s** | 179 | 172 | 182 |
> | **Avg. latency (s/request)** | 5.5 | 5.8 | 5.4 |
> | **Requests (avg. 8K, 0.97 GiB)** | 4 | 52 | 206 |
>
> **What We Observed**
>
> - When HBM is limited, both swapping and recompute mechanisms are triggered as expected.
> - HiFC successfully **hides SSD swap latency** through pipeline overlap, minimizing the impact on service performance.
> - Interestingly, HiFC delivered **5% higher throughput** than DRAM-based swapping, likely because DRAM contention can increase tail latency in some cases.
> - Overall, HiFC achieved **latency very close to DRAM** while supporting **4× more requests**.
>
> **Conclusion**
>
> Our evaluation suggests that HiFC’s design effectively prevents SSD latency from becoming a bottleneck. By leveraging vLLM’s scheduling and overlapping techniques, HiFC hides most of the swap cost and preserves low latency. These findings give us confidence that HiFC can be applied to **latency-sensitive inference scenarios** without introducing major concerns.
>
> ---
>
> **Q3.2.** Are the price ranges and other components used in the TCO analysis, particularly those in Table 1, sufficiently justified with citations given their critical role in supporting the paper’s cost reduction claims?
>
> **A3.2.** We apologize for not including explicit citations for the price ranges and components in our original TCO analysis. As cost and energy claims are central to the paper, we re-evaluated all parameters using updated market data. The original paper used prices from the **first half of 2025**, while the table below reflects data from the **second half of 2025** with added references for clarity.
>
> **Updated 3-years TCO Table**
>
> | **Metric** | **DRAM (DDR4-3200)** | **TLC SSD (GEN4)** | **HiFC (pSLC SSD, GEN4)** |
> | --- | --- | --- | --- |
> | **Capacity (GiB)** | 128 | 1920 | 1024 |
> | **Price_per_GiB ($)** | 5.06 | 0.20 | 0.10 |
> | **CapEx ($)** | 647 | 384 | 102 |
> | **Power Draw (W)** | 24 | 8.2 | 5 |
> | **PUE (ratio)** | 1.3 | 1.3 | 1.3 |
> | **Energy Cost ($/kWh)** | 0.1 | 0.1 | 0.1 |
> | **3-year TCO ($)** | 729 | 465 | 120 |
> | Previous ×4.5 | ×6.1 | ×3.9 | 1.0 |
>
> **Reference**
>
> - Memory, SSD: Memory.NET / Amazon
> - Power cost, PUE: 2024 US Data Center Energy Usage Report
>
> **Key Findings**
>
> - The updated analysis shows that the **3-year TCO gap has widened**: DRAM is now **6.1×** more expensive than HiFC (previously 4.5×).
> - This increase is mainly due to **recent DRAM price hikes**, further strengthening the cost advantage of SSD-based HiFC.
> - All cited prices are derived from publicly available sources, ensuring reproducibility of the analysis.
>
> **Conclusion**
>
> We regret not including these references in the original submission. The updated data confirms that our TCO claims remain valid and that the cost benefits of HiFC have become even more pronounced in the current market environment. This reinforces the paper’s main argument that HiFC provides **a practical and economically significant solution** for large-scale LLM inference deployments.
>
> ---
>
> **Q3.3.** What are the potential systems engineering challenges in maintaining SSD-based systems, particularly regarding increased failure rates?
>
> **A3.3.** We thank the reviewer for raising concerns about maintaining SSD-based systems. SSDs in HiFC are managed like standard GPU server drives with **NVMe logs** for monitoring, and failures have minimal impact as drives are easily replaced. Using **pSLC regions**, HiFC also ensures long SSD lifespan.
>
> **NVMe Error Management and Monitoring Technologies**
>
> - **SMART Log**: Shows health data (temperature, errors, remaining life) for early failure detection.
> - **Error Log**: Records I/O and PCIe errors to identify and troubleshoot faults.
> - **Self-Test Log**: Runs diagnostics to detect hidden issues before they impact performance.
> - **Telemetry Log**: Collects detailed metrics (I/O, wear) for tuning and reliability checks.
>
> **Conclusion**
>
> HiFC adds no extra risk beyond standard SSD-managed GPU servers. Using NVMe monitoring and quick drive replacement, **system stability is maintained**, and SSD-related engineering challenges remain minimal.
>
> ---
>
> **Q3.4.** Are the price trends for these technologies (DRAM versus SSD) such that the TCO benefits from this method will shrink, persist, or grow over the coming years?
>
> **A3.4.** We apologize for not including explicit citations for the price ranges and components in our original TCO analysis. We appreciate the reviewer’s observation, as cost and energy claims are central to our work and require clear justification.
>
> For this rebuttal, we have re-examined **current market prices (H2 2025)** and projected future trends for GPUs, DRAM, and SSDs. The updated values confirm that the **cost gap between DRAM and SSD has widened**, further strengthening HiFC’s economic advantage.
>
> **Projected 3-Year Price and TCO Trends (2025–2027)**
>
> | **Component** | **2025 (Current)** | **2026 (Projection)** | **2027 (Projection)** | **3-Year TCO Trend** |
> | --- | --- | --- | --- | --- |
> | **GPU (GB200)** | $30K – $35K | $25K – $30K | $20K – $25K | Remains high due to demand for cutting-edge AI training |
> | **DDR5 DRAM** (128 GiB) | $1,050 – $1,200  | $900 – $1,000 | $800 – $950 | **Decrease** as DDR5 adoption grows |
> | **Data center SSD** (1.92 TB) | $350 – $420  | $300 – $350 | $250 – $300 | **Steady decline** with NAND oversupply |
> | **pSLC SSD** (1 TB) | $95 – $120  | $85 – $110 | $75 – $100 | Remains cost-effective for HiFC |
> | **High-Perf. SSD** | $3,000+  | $2,000 – $2,500 | $1,500 – $2,000  | **Decrease** as GPU-direct SSD adoption grows |
>
> Thanks to the reviewer’s question, we re-examined DRAM/SSD price trends and found **SSDs’ TCO advantage has increased due to rising DRAM costs**. While GPUs like **Blackwell** require expensive **DDR5**, HiFC uses SSDs only for temporary KV cache, avoiding high-cost memory. Future high-performance SSDs will further enhance offloading, and HiFC is already compatible via GDS.
>
> **Conclusion:**
>
> The updated market data confirms that HiFC’s **TCO benefits are not only valid but are likely to become even more significant** over the coming years, reinforcing its practical value for large-scale LLM deployments.
>
> ---
>
> **Q3.5.** Are there any sustainability concerns from repeatedly burning through flash drives (even if this has lower TCO)?
>
> **A3.5.** We appreciate the reviewer’s question on SSD sustainability, as endurance affects both cost and environmental factors. HiFC ensures **long-term reliability** by using only the **pSLC region**, which has over **10×** the endurance of TLC NAND. In our 1 TB SSD, the 200 GiB pSLC region supports **6,000 TB writes**—**8×** the rated 750 TBW. HiFC also uses **sequential I/O** to reduce internal GC operations, further extending lifespan.
>
> **Experimental Setup**
>
> - GPU: A100 (80GB)
> - SSD for HiFC: 1TB (pSLC 200 GiB)
> - Datasets: GovReport (avg. 8.7K)
> - Output length:  1K
>
> **Predicted SSD Lifetime Based on Testing and Lifetime Calculation Formula**
>
> - Total KV written per day (TiB) = KV size per test (GiB) × Tests per day
> - Lifetime (years) = pSLC TBW (TiB) / (Total KV Written per Day (TiB) × 365)
> - **Lifetime** (years) = 6000 / (1.98 × 365) ≈ 8.3
>
> | **Metric** | **HiFC**  |
> | --- | --- |
> | **Swap-out count (writes)** | 28 |
> | **Avg. KV size (MiB)** (8.7K length) | 984 |
> | **KV size per test (GiB)** | 26 |
> | **Total test time (s)** | 1,097 |
> | **Tests per day** | 78 |
> | **Total KV written per day (TiB)** | 1.98 |
> | **pSLC TBW** | 6,000 |
> | **Predicted Lifetime (years)** | **8.3** |
>
> **Key Findings**
>
> - Under our test conditions, HiFC’s SSD would last **up to 8.3 years**, far exceeding typical service lifetimes.
> - For comparison, reports such as **Tom’s Hardware** estimate the expected service life of GPUs used for LLM inference at **around 3 years**.
> - This means the SSD would only need to be replaced **once** while the GPU may undergo **three replacements** in the same period.
>
> ---
>
> We have implemented these features and will release the full code and results on GitHub before the camera-ready version.
>
> We appreciate the reviewers’ time and consideration in reviewing our rebuttal.

---

> > ### Comment · Reviewer_bZj1 · 2025-08-03
> >
> > Thank you for the detailed response. The author rebuttal has addressed each of the concerns I raised as well as the questions I had. As such, I will stick with my original rating (Accept).

---

> > > ### Author Response · Authors · 2025-08-04
> > >
> > > Dear Reviewer bZj1,
> > >
> > > We thank the reviewer for taking the time to provide such a careful and thoughtful review of our work. The rebuttal period was particularly valuable, as it allowed us to reflect on the price trends of memory and storage as well as their impact on TCO. We greatly appreciate the insightful comments, and we will make sure to incorporate them into the final version of the paper.
> > >
> > > Sincerely,
> > >
> > > Authors of Paper # 12452

---

### Official Review · Reviewer_9KLT · 2025-07-05

**Clarity:** 3
**Significance:** 2
**Originality:** 3
**Rating:** 3
**Confidence:** 4

**Summary:**

This paper proposes HiFC, a novel DRAM-free approach for managing KV cache in large language model inference by directly swapping KV cache blocks between GPU memory and NVMe SSD storage. The main contribution of this paper is achieving comparable throughput to DRAM-based approaches while reducing total cost of ownership (TCO).

**Questions:**

See above

**Ethical Concerns:**

["NO or VERY MINOR ethics concerns only"]

**Final Justification:**

I think during the rebuttal the authors couldn't address the concerns I raised limited scope + the cost concern I raised. Therefore, I decided to keep the original rating.

**Limitations:**

yes.

**Quality:**

2

**Strengths And Weaknesses:**

Strengths:
- The DRAM-free design is innovative, bypassing the traditional CPU-DRAM data path.
- A direct GPU-to-SSD data path is a significant architectural advancement.
- The system leverages GPU Direct Storage, which sees the long-context problem from a system perspective, helping the community see the problem from another perspective.
- Integration with vLLM makes HiFC practical.

Limitations and Weaknesses:
- Limited Experimental Scope: Offloaded-based KV cache management is mainly designed to extend the memory capacity. We see no comparison with other offloaded-based KV cache management with a DRAM baseline. Would offloading to SSDs maintain a larger context/more KV-cache than offloading to DRAM? That is an intriguing question this paper didn't answer.
- The main contribution of this paper is reducing the total cost of ownership (TCO) by 4.5× over three years. We buy into this claim point, and we need to point out HiFC is working on a hybrid system with GPU/DRAM/SSD. And in most cases of cloud renting, the cost of owning a GPU is ten to a hundred times higher than the cost of owning DRAM or SSD. For example, ownership of 3-year DRAM costs \\$ 614 but ownership of 1-month A100 would cost over \$1000. Therefore, saving 4.5× ownership of KV-cache management would be negligible compared with the total cost of ownership of the GPU.

Overall, we think the DRAM-free design is innovative, but also believe in the design principle “lex parsimoniae”. Specifically, the HiFC adds an additional entity without performance gain, and TCO can be covered by the cost of GPU ownership. My biggest concern of the work is this is a practical / system contribution to the community but I don't think the gain / setting is justified. Does this design have impact on other settings or address other practical problems?

---

> ### Author Rebuttal · Authors · 2025-07-31
>
> We appreciate the reviewer’s thorough assessment of our work and the constructive suggestions provided. Our detailed responses to the raised points are given below.
>
> **Q2.1.** Is the paper missing a fair comparison with other DRAM-based offloading methods (FlexGen, DeepSpeed, etc.)?
>
> **A2.1.** We thank the reviewer for raising the concern about the lack of explicit comparison with other DRAM-based offloading methods. Existing offload-based frameworks, including **FlexGen**[6] and **DeepSpeed**[9], rely on host DRAM to store KV cache, but their behavior differs from that of **vLLM**[5].
>
> **Offloading vs. Swapping: Key Differences**
>
> | Aspect | **Offloading (FlexGen, DeepSpeed)** | **Swapping (vLLM, HiFC)** |
> | --- | --- | --- |
> | **Primary Storage** | Host DRAM  / SSD(always used for KV cache) | GPU HBM (primary) + DRAM / SSD (only when HBM is insufficient) |
> | **When Data Moves** | KV cache always stored in DRAM, fetched on demand | KV cache stored in HBM until space runs out, then swapped out |
> | **Performance Impact** | High I/O overhead due to frequent host DRAM access | Latency mostly hidden by scheduler (decoding continues) |
> | **HiFC Enhancement** | Not applicable | Uses SSD + GDS to replace DRAM, reducing cost and expanding capacity |
> - **FlexGen** and **DeepSpeed** **store KV cache in host DRAM** continuously and **fetch it directly during attention.**
> - **vLLM** keeps KV cache in GPU HBM and **only swaps out to host DRAM** when necessary, hiding most latency through its scheduler.
>
> **We would like to emphasize that a direct, one-to-one comparison between the offloading and swapping techniques is inherently difficult, since they rely on different inference frameworks.** Even when using the same GPU only, the performance of vLLM and DeepSpeed can differ substantially, as each framework employs fundamentally different methods for attention processing and scheduling. **Therefore, we compared the DRAM‐based swapping technique implemented in vLLM with our HiFC method, which is also implemented in vLLM.** By keeping the framework constant, **we isolated the effect of storage type (DRAM vs. SSD), eliminating confounding factors** and clearly demonstrating the benefits of HiFC, which builds on vLLM while replacing DRAM with SSDs to achieve comparable latency at a significantly lower cost.
>
> ---
>
> **Q2.2.** Does HiFC enable maintaining larger KV cache and context than DRAM-based offloading?
>
> **A2.2.** We thank the reviewer for this important question. **Yes, HiFC enables maintaining a significantly larger KV cache and supports longer contexts compared to DRAM-based offloading.** HiFC builds on vLLM’s swapping mechanism but replaces host DRAM with SSDs, which greatly expands storage capacity while keeping latency low. Under the **same TCO as DRAM, HiFC can store over 6.2× more KV cache.**
>
> **Memory Scaling for KV cache**
>
> Table below compares request capacity and relative TCO across different memory configurations.
>
> | Item | Dataset Name / Data Size |
> | --- | --- |
> | **Model** | DeepSeek-R1-Distill-Llama-8B |
> | **Datasets** | GovReport (avg. 8.7K) |
> | **Avg. KV cache size**  | 984 MiB |
> - **Requests = Memory Size / avg. KV cache size**
>
> | **Multi setting** | **Memory Size** | **Requests (avg. 8K)** | **Requests Ratio** | **3-year TCO (relative)** |
> | --- | --- | --- | --- | --- |
> | **GPU HBM** | 80 GiB | 82 | 0.6× | – |
> | **Host DRAM** | 128 GiB | 132 | 1 | 4.5× |
> | **GPU - SSD x 1** | 1 TiB (pSLC 200 GiB) | 206 | **1.5×** | 1× |
> | **GPU - SSD x 2** | 2 TiB (pSLC 200 GiB ×2) | 412 | **3.1×** | 2× |
> | **GPU - SSD x 4** | 4 TiB (pSLC 200 GiB ×4) | 824 | **6.2×** | 4× |
>
> ---
>
> - For the cost of DRAM, operators can provision **four SSDs**, enabling up to **6.2× more requests compared to DRAM**
> - The pSLC region of a 1 TiB SSD (200 GiB) can be configured as a **800 GiB pSLC** with multiple SSDs in **RAID0**.
> - With vLLM’s flexible allocation, HiFC fully exploits this expanded Flash cache without requiring model modifications.
>
> In practice, configuring **four SSDs** in HW/SW RAID0 and registering the Flash cache file in HiFC enables **seamless scaling**, allowing service providers to handle **longer contexts** and **more requests** while maintaining **significantly lower costs** than DRAM-based offloading.
> **Conclusion**
>
> These results demonstrate that **HiFC not only matches DRAM-based approaches in performance but also provides a much larger KV cache capacity at lower cost**, directly addressing the reviewer’s concerns on scalability and impact.
>
> ---
>
> **Q2.3.** Is the 4.5× TCO saving from HiFC significant when GPU ownership costs dominate the overall expenses in cloud environments?
>
> **A2.3.** We agree with the reviewer that in cloud environments, **GPU rental costs overshadow DRAM or SSD ownership costs**. Under such conditions, our 4.5× TCO savings for KV cache management may appear less significant compared to the high monthly cost of GPUs.
>
> However, HiFC targets a **different deployment model**: **LLM service providers operating on-premise GPU infrastructure**. In these environments, operators must optimize **memory expansion costs** rather than GPU rental fees when scaling to handle larger KV caches and longer contexts.
>
> - **For cloud users**, scaling typically means **renting more GPU instances**, where costs are dictated by the provider’s pricing model.
> - **For on-premise providers**, scaling can be achieved by **adding low-cost SSDs** to existing servers, avoiding expensive GPU upgrades.
> - SSDs are **simple to deploy**, can **reuse existing hardware**, and enable **quick scaling without downtime**.
>
> In this context, HiFC provides a valuable alternative: rather than purchasing an additional **$14,000 A100 GPU**, an operator can **expand KV cache capacity** with a **$100 SSD**, achieving substantial scalability at minimal cost.
>
> **Cost Comparison (3-Year Ownership vs. Cloud Rental)**
>
> | **Resource** | **Ownership (3-Year)** | **Cloud Rental (1 Month)** | **Remark** |
> | --- | --- | --- | --- |
> | **GPU (A100)** | ≈ **$14,000** | ≈ **$1,000+** | Cloud rental can exceed $12,000/year |
> | **DRAM (128GB ECC DDR4)** | ≈ **$600** | Included in GPU instance | Ownership cost is relatively low |
> | **SSD (1TB pSLC)** | ≈ **$100** | Included in GPU instance | Easy to add/scale in on-premise setups |
>
> **Key Takeaways**
>
> - **For cloud services**, GPU rental dominates total costs, making memory-related savings less impactful.
> - **For infrastructure owners**, **HiFC** provides a **highly cost-efficient scaling method**, replacing costly GPU upgrades with inexpensive SSDs.
> - HiFC allows **seamless KV cache expansion** within vLLM without architectural changes, making it a **practical and economically meaningful solution** beyond the context of cloud rentals.
>
> **Conclusion**
>
> While HiFC’s TCO savings may be negligible in cloud rental scenarios, **it becomes highly significant for service providers managing their own GPU infrastructure**. The ability to expand memory using commodity SSDs instead of costly GPUs provides a compelling advantage in real-world deployments.
>
> ---
>
> **Q2.4.** Does the DRAM-free HiFC design deliver meaningful benefits and practical impact beyond the added SSD component, given that GPU ownership costs dominate the overall TCO?
>
> **A2.4.** We thank the reviewer for this important question. **Yes, HiFC provides meaningful benefits and practical impact that go far beyond simply adding an SSD, even when GPU ownership costs dominate overall TCO.** HiFC was specifically designed to address the high cost and limited capacity of DRAM in KV cache management, offering a scalable and cost-effective solution without compromising performance.
>
>  **Summary of HiFC’s Contributions**
>
> | **Aspect** | **HiFC’s Contribution** |
> | --- | --- |
> | **Cost Efficiency Beyond GPU Ownership** | While GPU rental costs dominate in cloud settings, HiFC targets **on-premise service providers**, delivering **4.5× lower KV cache management cost** than DRAM solutions and avoiding costly GPU upgrades. |
> | **Practical System Benefits** | HiFC enables operators to **expand KV cache capacity using inexpensive SSDs**, effectively removing memory bottlenecks in long-context inference. |
> | **Applicability to Other Settings** | HiFC supports **multi-GPU and multi-SSD configurations** with near-linear throughput scaling. It integrates seamlessly with **vLLM** without requiring model changes, and installation details are provided in ***HiFC_README.pdf*** within the supplementary material. |
>
>  **Additional Remarks**
>
> Beyond cost savings, **HiFC introduces a new design principle for KV cache scaling**, utilizing commodity SSDs to overcome memory limits without architectural modifications. It extends the widely adopted **vLLM** framework, making it **immediately usable** by both researchers and practitioners. The full implementation will be released as open source to encourage adoption.
>
> In real-world deployments, instead of upgrading to an additional $14,000 A100 GPU (80 GiB), **operators can simply add a $100 SSD (1 TiB), expanding KV cache capacity by up to 12.8×** with minimal modifications to their systems.
>
>  **Conclusion**
>
> These results show that HiFC is **not just a DRAM replacement using SSDs**, but a **practical, scalable, and cost-efficient solution** to the KV cache scalability problem. We believe HiFC makes a **meaningful contribution** to enabling **economical, high-performance** long-context LLM inference, directly addressing the reviewer’s concerns regarding its impact and applicability.
>
> ---
>
> We have implemented these features and will release the full code and results on GitHub before the camera-ready version.
>
> We appreciate the reviewers’ time and consideration in reviewing our rebuttal.

---

> > ### Comment · Reviewer_9KLT · 2025-08-04
> > **Additional concerns**
> >
> > Thank you for the clarifications. I now see the proposed system differs from traditional offloaded LLM inference systems. However, I still have several critical concerns regarding the claimed cost efficiency:
> >
> > - Scalability and Response Time:  The authors claim the system can "scale to handle larger KV caches and longer contexts" and "achieve substantial scalability at minimal cost." However, in practice, the ability to store larger KV caches contributes only marginally to overall inference performance. As user load increases or context length grows, deployment inevitably requires scaling up GPU resource, either by upgrading GPUs or increasing their quantity, to maintain acceptable response times. Without such scaling, the latency would become unacceptable. Moreover, the batch sizes reported in Rebuttal A2.2 (e.g., serving 136 or more requests on a single A100) seem far from realistic for practical use cases. Additionally, many serving systems employ alternative techniques beyond page swapping, such as KV cache recomputation, to manage memory and latency.
> > - TCO and Energy Savings: The paper cites a 4.5× total cost of ownership (TCO) reduction, primarily based on energy savings. However, this comparison overlooks a fundamental disparity: the energy consumption of DRAM, SSDs, and HiFC (ranging from ~5W to ~64W) is negligible compared to that of GPUs—where a single A100 draws around 400W and an H100 about 700W. Even in on-premise deployments, the GPU remains the dominant contributor to energy costs. The rebuttal does not adequately address this imbalance, and the claimed TCO improvements appear overstated.
> > Related Work and Prior Art: The paper omits discussion of several relevant and recent works that also address long-context serving via DRAM/SSD-based approaches, beyond the page swapping in vLLM, such as AttentionStore [1] and Mooncake [2].
> >
> > [1] Gao, Bin, et al. "{Cost-Efficient} large language model serving for multi-turn conversations with {CachedAttention}." 2024 USENIX Annual Technical Conference (USENIX ATC 24). 2024.
> > [2] Qin, Ruoyu, et al. "Mooncake: A kvcache-centric disaggregated architecture for llm serving." arXiv preprint arXiv:2407.00079 (2024).

---

### Official Review · Reviewer_Jq6w · 2025-07-20

**Clarity:** 3
**Significance:** 3
**Originality:** 4
**Rating:** 5
**Confidence:** 3

**Summary:**

This paper introduces a system-level optimization to efficiently store KV cache. The authors propose to bypass DRAM, and do direct GPU–SSD swapping using fast SSD technology and NVIDIA’s GPU-Direct Storage. Specifically, they use pSLC regions of SSDs and seemelessly integrate with vLLM using a block-based memory allocator and swap engine. The authors present comprehensive results to support the efficacy of the proposed method.

**Questions:**

NA

**Ethical Concerns:**

["NO or VERY MINOR ethics concerns only"]

**Final Justification:**

I maintain my original score

**Limitations:**

Yes

**Quality:**

3

**Strengths And Weaknesses:**

Strengths
- Motivation is clear. DRAM offloading is expensive and leads to extra latency overhead.
- Idea is novel. The paper leverages specific regions of the SSD to optimize the cost, ensuring DRAM free SSD swapping.
- Results show upto 4.5x TCO reduction in 3 years.

Weaknesses
- It is not clear how the system will scale with multi-GPU system
- Results reported on only one model. It will be helpful if authors can share results on other LLMs.

---

> ### Author Rebuttal · Authors · 2025-07-31
>
> We appreciate the reviewer’s thorough assessment of our work and the constructive suggestions provided. Our detailed responses to the raised points are given below.
>
> **Q1.1.** How will the system scale in a multi-GPU environment?
>
> **A1.1.** We thank the reviewer for raising the important question about HiFC’s scalability in multi-GPU environments.
> To address this concern, we have extended **HiFC to operate efficiently on multi-GPU and multi-SSD configurations** and conducted additional evaluations to verify its behavior under these setups.
>
> HiFC supports three deployment topologies:
>
> - **1:1 (GPU:SSD)** – baseline configuration commonly used in workstation setups
> - **N:1** – **multiple GPUs sharing a single SSD** to minimize storage cost
> - **N:N** – **multiple GPUs connected to multiple SSDs** (with RAID) to maximize both performance and bandwidth
>
> **Experimental Configuration**
>
> | Item | Specification |
> | --- | --- |
> | **GPU** | A100 (80GiB) × 2 |
> | **HBM KV cache size** | 4 GiB (starvation condition) |
> | **DRAM KV cache size** | 50 GiB |
> | **SSD for HiFC** | 1TB (pSLC 200 GiB) × 2 |
> | **Model** | DeepSeek-R1-Distill-Llama-8B |
> | **Dataset** | THUDM/LongBench/gov_report_e (avg. 8K length) |
>
> **Extended Evaluation Results**
>
> | Setting | DRAM / pSLC Size | TPS (tokens/s) | Requests (avg. 8K) |
> | --- | --- | --- | --- |
> | **GPU:DRAM** | 50 GiB | 172 | 51 |
> | **GPU:SSD = 1:1** | 200 GiB | **182** | 206 |
> | **GPU:SSD = 2:1** | 200 GiB | **367** | 206 (103 + 103) |
> | **GPU:SSD = 2:2** | 400 GiB | **364** | 412 (206 + 206) |
>
> **Observations**
>
> - The **1:1** configuration already provides **5% higher TPS** than DRAM while expanding KV cache capacity.
> - In the **2:1** configuration, **throughput nearly doubles (~2×)** as the number of GPUs increases because swap I/O collisions on the SSD are rare.
> - In the **2:2** configuration, **throughput also achieves a near-2×** increase while additionally supporting a higher number of requests.
> - In multi-GPU configurations, **each GPU is allocated the same pSLC region for KV cache**, following vLLM’s cache management policy.
> - The maximum request capacity scales with the number of SSDs. Choose SSD count based on service scale.
>
> **Conclusion**
>
> These experiments demonstrate that **HiFC scales robustly in multi-GPU and multi-SSD environments**, achieving near-linear throughput scaling without sacrificing memory capacity.
>
> ---
>
> **Q1.2.** Can the authors provide results on other LLMs to demonstrate broader applicability?
>
> **A1.2.** We thank the reviewer for asking about HiFC’s applicability to different LLMs and workloads. HiFC was developed with practical deployment scenarios in mind and has been tested with diverse datasets during development. In the original paper, we intentionally used a fixed model and input/output lengths to provide a fair and controlled comparison against DRAM-based swapping. **To further demonstrate generality, we conducted additional evaluations using several widely adopted open-source models (DeepSeek-Llama-8B, DeepSeek-Qwen-14B, and Mistral-7B**) across datasets with context lengths ranging from a few hundred tokens to over 18K tokens. All experiments were performed under the same GPU/SSD configuration as in the paper.
>
> **Experimental Setup**
>
> | Item | Specification |
> | --- | --- |
> | **GPU** | A100 (80GB) |
> | **HBM KV cache size** | 4 GiB (starvation condition) |
> | **KV swap memory** | DRAM (DDR4) / SSD (GEN4) |
> | **Datasets** | **Qasper** (avg. 3.6K), **GovReport** (avg. 8.7K), **NarrativeQA** (avg. 18.4K) |
> | **Block size** | 64 tokens |
> | **Output length** | 1K |
>
> **Extended Evaluation Results**
>
> | Model | KV Swap | Qasper | GovReport | NarrativeQA | Observation |
> | --- | --- | --- | --- | --- | --- |
> | **DS-Llama-8B** | DRAM | 302.0 | 172.0 | 95.1 |  |
> |  | **HiFC** | **301.3** | **182.3** | **95.8** | Avg. diff. ~2% |
> | **DS-Qwen-14B** | DRAM | 176.4 | 100.9 | 46.4 |  |
> |  | **HiFC** | **175.3** | **103.5** | **46.6** | Avg. diff. ~1% |
> | **Mistral-7B** | DRAM | 281.0 | 163.0 | 416.1* | NarrativeQA limited to <32K |
> |  | **HiFC** | **275.2** | **162.0** | **406.8*** | Avg. diff. ~2% |
>
> **Key Observations**
>
> - Across all tested models and datasets, **the performance gap between DRAM and HiFC remains within ~2%**, confirming negligible overhead.
> - These findings align with the results presented for DS-Qwen-32B in the paper.
> - The 32K context limit of Mistral affected some NarrativeQA requests, but the overall trend remained consistent.
>
> **Conclusion**
>
> The additional experiments demonstrate that **HiFC consistently delivers DRAM-comparable throughput** across multiple LLM architectures and dynamic context lengths, while achieving a **4.5× reduction in total cost of ownership** by replacing expensive DRAM with a low-cost pSLC SSD. This confirms that HiFC is broadly applicable and effective for long-context LLM inference in realistic deployment scenarios.
>
> ---
>
> We have implemented these features and will release the full code and results on GitHub before the camera-ready version.
>
> We appreciate the reviewers’ time and consideration in reviewing our rebuttal.

---

### Note · Authors · 2025-08-11

We thank the reviewers and the area chair for their careful evaluation.

This work presents **HiFC**, a DRAM-free KV-cache swapping scheme that uses GPU-Direct Storage to transfer KV blocks directly between GPU and an SSD configured with a pSLC region. The design **avoids host-memory staging** and **preserves DRAM-level end-to-end throughput** under the evaluated long-context workloads.

To preclude ambiguity, the final version will explicitly state that our TCO figures refer to the **additional TCO for memory expansion**. As reported in the paper, substituting 128 GiB DRAM with a 1 TiB pSLC SSD yields a 4.5× reduction in three-year memory-related TCO while maintaining comparable throughput to DRAM-based swapping.

We believe these results make **HiFC a practical, cost-effective approach for scaling LLM serving without compromising stability**. We will release the HiFC code on GitHub and aim to maintain compatibility with recent vLLM releases.

Sincerely,

Authors of Paper #12452

---

### Decision · Program_Chairs · 2025-09-17

**Decision:**

Accept (poster)

**Comment:**

This paper presents a system-level optimization for efficient KV cache storage and received mixed reviews. Three reviewers were very positive, while one expressed concerns about the practical applicability of the approach. The AC acknowledged these limitations but considered the work interesting and worthy of publication at NeurIPS, ultimately recommending acceptance.